# A modified BPaL regimen for tuberculosis treatment replaces linezolid with inhaled spectinamides

**Malik Zohaib Ali[1,2,3], Taru S Dutt[1,2], Amy MacNeill[2], Amanda Walz[1,2], Camron Pearce[1,2,3], Ha Lam[1,2], Jamie S Philp[1,2], Johnathan Patterson[1,2], Marcela Henao-Tamayo[1,2], Richard Lee[4], Jiuyu Liu[4], Gregory T Robertson[1,2], Anthony J Hickey[5], Bernd Meibohm[6], Mercedes Gonzalez Juarrero[1,2]***

[1]Mycobacteria Research Laboratories, Colorado State University, Fort Collins, United States; [2]Microbiology, Immunology and Pathology, Colorado State University, Fort Collins, United States; [3]Program in Cell & Molecular Biology, Colorado State University, Fort Collins, United States; [4]Department of Chemical Biology and Therapeutics, St. Jude Children's Research Hospital, Memphis, United States; [5]Technology Advancement and Commercialization, RTI International, Research Triangle Park, United States; [6]Department of Pharmaceutical Sciences, University of Tennessee Health Science Center, Memphis, United States

**\*For correspondence:** mercedes.gonzalez-juarrero@colostate.edu

**Competing interest:** The authors declare that no competing interests exist.

**Preprint posted** 16 November 2023 **Sent for Review** 23 January 2024 **Reviewed preprint posted** 12 April 2024 **Reviewed preprint revised** 14 August 2024 **Version of Record published** 08 October 2024

**Abstract** The Nix-TB clinical trial evaluated a new 6 month regimen containing three oral drugs; bedaquiline (B), pretomanid (Pa), and linezolid (L) (BPaL regimen) for the treatment of tuberculosis (TB). This regimen achieved remarkable results as almost 90% of the multidrug-resistant or extensively drug-resistant TB participants were cured but many patients also developed severe adverse events (AEs). The AEs were associated with the long-term administration of the protein synthesis inhibitor linezolid. Spectinamide 1599 is also a protein synthesis inhibitor of *Mycobacterium tuberculosis* with an excellent safety profile, but it lacks oral bioavailability. Here, we propose to replace L in the BPaL regimen with spectinamide (S) administered via inhalation and we demonstrate that inhaled spectinamide 1599, combined with BPa —BPaS regimen—has similar efficacy to that of the BPaL regimen while simultaneously avoiding the L-associated AEs. The BPaL and BPaS regimens were compared in the BALB/c and C3HeB/FeJ murine chronic TB efficacy models. After 4-weeks of treatment, both regimens promoted equivalent bactericidal effects in both TB murine models. However, treatment with BPaL resulted in significant weight loss and the complete blood count suggested the development of anemia. These effects were not similarly observed in mice treated with BPaS. BPaL and BPa, but not the BPaS treatment, also decreased myeloid to erythroid ratio suggesting the S in the BPaS regimen was able to recover this effect. Moreover, the BPaL also increased concentration of proinflammatory cytokines in bone marrow compared to mice receiving BPaS regimen. These combined data suggest that inhaled spectinamide 1599 combined with BPa is an effective TB regimen without L-associated AEs.

## eLife assessment

In this **useful** study, the authors report the efficacy, hematological effects, and inflammatory response of the BPaL regimen (containing bedaquiline, pretomanid, and linezolid) compared to a variation in which Linezolid is replaced with the preclinical development candidate spectinamide 1599, administered by inhalation in tuberculosis-infected mice. The authors provide **convincing** evidence that supports the replacement of Linezolid in the current standard of care for

drug-resistant tuberculosis. The work will be of interest to those studying tuberculosis treatment regimens.

## Introduction

TB remains one of the leading causes of death initiated by an infectious agent. In 2021, the World Health Organization reported 10.6 million new TB cases worldwide, and among those, 450,000 cases were also diagnosed as multidrug-resistant (MDR) or extensiverly drug-resistant (XDR) TB (*Campbell et al., 2022*). Treatment of MDR- and XDR-TB patients is lengthy and is often poorly tolerated due to significant associated side effects (*Zhang et al., 2021*).

Recently, a 6 month novel treatment regimen of three oral drugs: bedaquiline (B), pretomanid (Pa), and linezolid (L) referred to as the BPaL regimen was approved. Preclinical studies demonstrated the better efficacy of BPaL regimen for drug-sensitive TB compared to the standard TB chemotherapy (*Campbell et al., 2022*; *Williams et al., 2012*; *Tasneen et al., 2016*) and thereafter, the BPaL regimen was tested in the Nix-TB clinical trial conducted in South Africa (*Conradie et al., 2020*). This trial enrolled patients with XDR-TB and treatment-intolerant or non-responsive MDR-TB, including HIV-positive patients with a CD4 count of 50 or higher. The results were remarkable as 95 out of 107 patients were cured though many patients had a high rate of treatment-associated AEs.

The long-term administration of linezolid (an oxazolidinone antibiotic) was likely the causative agent resulting in bone marrow myelosuppression (48%), peripheral neuropathy, optic neuritis (81%), and anemia (37%) in patients treated with the BPaL regimen (*Conradie et al., 2020*). Subsequently, the ZeNix trial adjusted the BPaL regimen to a linezolid dose of 600 mg. This trial also had remarkable results; it cured 84–91% of patients (9–26 weeks of therapy, respectively) and resulted in fewer AEs than those observed in the Nix-TB trial (*Conradie et al., 2022*). Apart from AEs, several studies have also raised awareness of high doses of linezolid leading to the development of linezolid-resistant Mtb (*TB-Alliance, 2022*; *Jackson, 2007*; *Richter et al., 2007*). At present, the TB-drug development field is working to modify the BPaL regimen to maintain or improve its efficacy while diminishing treatment-associated AEs (*Li et al., 2023*).

Spectinomycin, an aminocyclitol antibiotic, is a broad-spectrum antibiotic used mainly for the treatment of *Neisseria gonorrhoeae* (*Holloway, 1982*). It inhibits bacterial protein synthesis (*Kanchugal P and Selmer, 2020*) and has an acceptable safety profile with no known ototoxicity and nephrotoxicity (*Owusu et al., 2022*; *de Jager and van Altena, 2002*). The activity of spectinomycin against Mtb is very poor but its structural modification led to the development of a new series of semisynthetic analogs called spectinamides (*Kanchugal P and Selmer, 2020*; *Temrikar et al., 2023*; *Lee et al., 2014*). Spectinamides bind selectively with the bacterial 30 S ribosomes and importantly, unlike linezolid, they do not bind to the mitochondrial 30 S in mammalian cells. The latter represents a great advantage for reduced potential for side effects, such as ototoxicity and myeloid suppression that are commonly associated with other protein synthesis inhibitors such as amikacin and linezolid, respectively (*De Vriese et al., 2006*; *Modongo et al., 2015*). Spectinamides' potent anti-tubercular activity is attributed to its ability to evade drug efflux by Rv1258c major facilitator superfamily transporter present on the surface of Mtb (*Liu et al., 2017*; *Bruhn et al., 2015*). Spectinamides have shown excellent activity against MDR- and XDR-Mtb strains (*Lee et al., 2014*; *Robertson et al., 2017*) however, their poor oral availability has limited their usage to injectable forms.

One of the lead spectinamides, 1599, has demonstrated promising results *in vitro* and *in vivo* and was shown to lack cross-resistance with existing anti-TB drugs (*Lee et al., 2014*; *Liu et al., 2017*; *Bruhn et al., 2015*; *Robertson et al., 2017*; *Gonzalez-Juarrero et al., 2021*; *Wagh et al., 2021*). 1599, delivered subcutaneously, proved an effective partner agent when combined with rifampin and pyrazinamide and also with bedaquiline, pretomanid, or moxifloxacin in TB mouse efficacy models of increasing complexity (*Robertson et al., 2017*).

One of the limitations of using 1599 as an injectable is the potential risk for poor patient compliance (*Nirmal et al., 2021*) and direct administration of aerosolized antibiotics to the lungs has been studied for decades as an alternative to systemic drug administration via injection. Aerosolized administration of 1599 has been tested in preclinical *in vivo* studies using the liquid formulation of the drug. These studies have shown that inhaled 1599, used in monotherapy or in combination with pyrazinamide, is efficacious and well tolerated in murine TB efficacy models (*Boisson et al., 2014*;

*Rathi et al., 2019*). A comparative study assessing the biodistribution of the drug in relation to the administration route demonstrated that 1599 showed 48 times higher exposure in mouse lungs via inhalation compared to equivalent dosages administered by subcutaneous injection; the latter may explain the increased efficacy of this drug following intrapulmonary aerosol (*Gonzalez-Juarrero et al., 2021*; *Rathi et al., 2019*). Moreover, 1599 was shown to be amenable to dry powder formulation and delivery, suggesting a pathway to a more patient-friendly delivery system (*Hickey et al., 2013*). Therefore, in this study, we hypothesized that combining BPa with inhaled spectinamide 1599 (S) will maintain equivalent efficacy to the BPaL regimen while avoiding the accompanying toxicities that occurred with long-term BPaL administration to human MDR/XDR-TB patients. Based on the diversity of outcomes observed during human TB disease (*Gonzalez-Juarrero et al., 2021*) and as no single animal model recapitulates the wide spectrum of human TB pathology (*Singh and Gupta, 2018*), we chose the BALB/c and C3HeB/FeJ murine TB models. The BALB/c chronic TB model is representative of a long-term Mtb chronic infection that develops homogenous lung granulomatous lesions restraining the bacilli within intracellular compartments (*De Groote et al., 2011*) of macrophages and foamy macrophages. In contrast, low-dose aerosol Mtb infection of C3HeB/FeJ mice also results in a chronic infection, but their lungs exhibit a heterogenous spectrum of lesions including granulomas similar to those seen in BALB/c chronic TB model in addition to caseous necrotic lesions surrounded by a fibrotic rim (*Robertson et al., 2021*; *Irwin et al., 2015*; *Driver et al., 2012*). The caseum of these necrotic granulomas creates a hypoxic environment and contains abundant extracellular bacilli (*Irwin et al., 2015*; *Driver et al., 2012*; *Lanoix et al., 2015*) in a similar fashion to necrotic granulomas found in some human TB patients. It is believed that the fibrotic, necrotic, and hypoxic environment of these granulomas creates barriers to drug penetration, alters bacterial phenotype, and all together challenges therapeutic outcomes (*Harper et al., 2012*; *Irwin et al., 2014*; *Aly et al., 2006*; *Walter et al., 2023*). Therefore, to understand the implications of drug efficacy and drug-associated AEs in scenarios without (BALB/c) and with (C3HeB/FeJ) necrotic granulomas, both murine TB efficacy models were employed.

## Results

### Linezolid and spectinamide 1599 show similar efficacy in monotherapy

To compare the efficacy of L or S in monotherapy, Mtb-infected C3HeB/FeJ (n=7) and BALB/c (n=4–6) mice received L (administered 5/7 days per week orally at 100 mg/Kg) or S (administered 3/7 days per week on alternate days via intrapulmonary aerosol delivery at 100 mg/Kg and 50 mg/Kg, respectively) for 4 weeks. At the end of treatment, the animals were sacrificed, and the CFU in the lungs and spleen was enumerated (*Figure 1A–D*). Treatment of C3HeB/FeJ mice with L ($7.30\pm0.45$ $\log_{10}$) or S ($6.90\pm0.48$ $\log_{10}$) for 4 weeks resulted in an average of 0.51 and 0.91 $\log_{10}$ CFU reduction in the lungs respectively, compared to untreated (UnRx) control ($7.81\pm0.36$ $\log_{10}$) (*Figure 1A*). This difference failed to achieve statistical significance owing to the larger standard deviation associated with heterogenous advanced lung pathology observed in this model (*Rathi et al., 2019*; *Irwin et al., 2015*; *Driver et al., 2012*). In contrast, L ($4.49\pm0.10$ $\log_{10}$) treated C3HeB/FeJ mice had significantly lower spleen bacterial burden compared to UnRx ($5.43\pm0.27$ $\log_{10}$), with no significant difference between the L and S ($4.94\pm0.11$ $\log_{10}$) treatment arm (*Figure 1C*). In BALB/c mice, L or S treatment was found to promote a significant reduction in lung bacterial burden (0.83 and 0.77 $\log_{10}$, respectively) compared to UnRx or the vehicle-only controls (*Figure 1B*). There was no significant difference in lung bacterial burden after L or S treatment in BALB/c mice and there was no change in spleen bacterial burden compared to UnRx control (*Figure 1D*).

### Combination therapy with BPaL or BPaS has similar efficacy

We further tested and compared the efficacy of L or S when used in combination therapy with BPa. Mtb-infected C3HeB/FeJ mice were treated with either BPaL (B=25 mpk; Pa = 100 mg/Kg and L=100 mg/Kg all administered 5/7 a week via oral gavage) or BPaS (BPa as in BPaL and S=100 mg/Kg administered 3/7 a week on alternate days via intrapulmonary aerosol delivery) for 4 weeks. The comparative analysis from combined data of three independent studies is shown in *Figure 1E and G* (data from individual studies are shown in *Figure 1—figure supplements 1–3*). Two of the three studies contained an extra group of BPa-treated mice as a reference control. Compared to UnRx ($7.48\pm0.12$

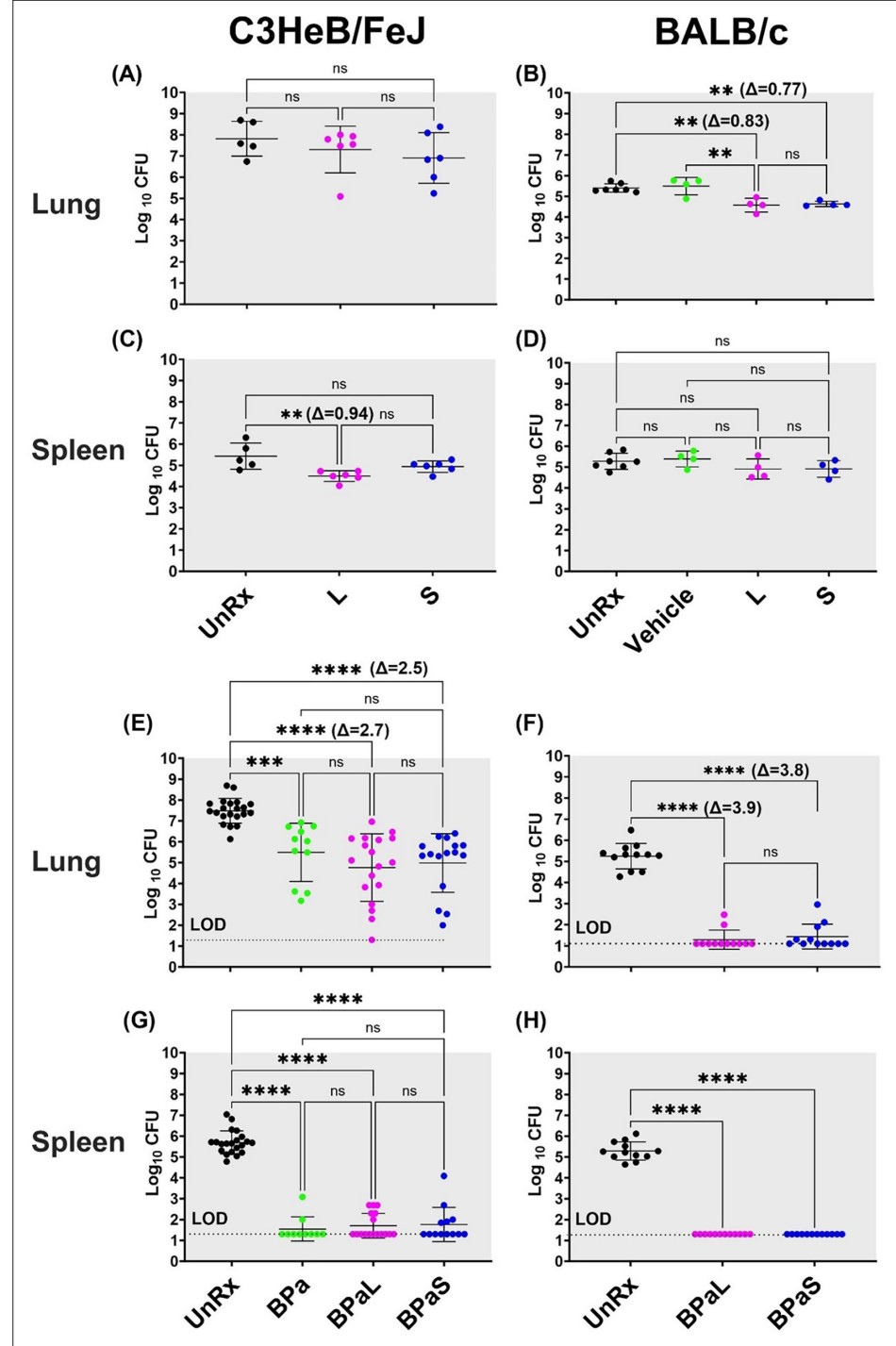

**Figure 1.** Bactericidal effect of BPaL and BPaS in TB mouse models after 4-weeks of treatment. BALB/c and C3HeB/FeJ female mice were chronically infected with a low dose aerosol infection of Mtb Erdman strain to deliver ~75 and ~100 bacilli respectively. Post-infection, BALB/c and C3HeB/FeJ mice were rested for 4 and 8-9 weeks respectively until they were randomly assigned to the study groups. The mice were treated with monotherapy of linezolid or spectinamide 1599 or combination therapy of BPaL, BPa or BPaS for 4 weeks. Bedaquiline (**B**), pretomanid (Pa) and linezolid (L) were administered at 25, 100 and 100 mg/kg respectively by oral gavage for 5/7 a week while spectinamide 1599 at 50 and 100 mg/kg in BALB/c and C3HeB/FeJ TB models respectively for 3/7 a week on the alternate days via intrapulmonary aerosol delivery. On the third day of the last treatment, the mice were euthanized, and their lungs and spleen were collected. The organs were homogenized, serially diluted and plated on 7H11 agar with 4% charcoal (to avoid drug carry-over effect) to determine bacterial burden

*Figure 1 continued on next page*

*Figure 1 continued*

in the form of colony forming units (CFU) in each sample. CFU were enumerated after 4-6 weeks of incubation at 37 °C and expressed as $\log_{10}$.C3HeB/FeJ and BALB/c TB models showing efficacy of monotherapy (**A-D**) and combination therapy (**E-H**). The C3HeB/FeJ graphs (**E, G**) represent the pooled data from three independent studies (n=3-8, *Figure 1—figure supplements 1–3*) and two of the three studies contained BPa as a reference control. The BALB/c graphs (**F, H**) represent the pooled data from two independent studies (n=5, *Figure 1—figure supplements 8 and 9*). Statistical significance was calculated using one-way ANOVA with Tukey's multiple comparison test, $p < 0.05$ was considered significant and ** = $p<0.001$, *** = $p<0.0001$, **** = $P<0.0001$. UnRx = untreated, L = linezolid, S = spectinamide 1599, LOD: limit of detection.

The online version of this article includes the following source data and figure supplement(s) for figure 1:

**Source data 1.** cfu of monotherapy in C3HeB/FeJ mice.

**Source data 2.** cfu of monotherapy in BALB/c mice.

**Source data 3.** cfu of combination therapy in C3HeB/FeJ mice (combined data).

**Source data 4.** cfu of combination therapy in BALB/c mice (combined data).

**Figure supplement 1.** C3HeB/FeJ Study 1, bacterial burden (CFU) in Mycobacterium tuberculosis infected C3HeB/FeJ mice treated with BPaL and BPaS regimen for 4 weeks CFU data.

**Figure supplement 1—source data 1.** Numerical values of CFU in *Figure 1—figure supplement 1*.

**Figure supplement 2.** C3HeB/FeJ Study 2, bacterial burden (CFU) in *Mycobacterium tuberculosis* infected C3HeB/FeJ mice treated with BPa, BPaL and BPaS regimen for 4 weeks CFU data.

**Figure supplement 2—source data 1.** Numerical values of CFU for *Figure 1—figure supplement 2*.

**Figure supplement 3.** C3HeB/FeJ Study 3, bacterial burden (CFU) in Mycobacterium tuberculosis infected C3HeB/FeJ mice treated with BPa, BPaL and BPaS regimen for 4 weeks.

**Figure supplement 3—source data 1.** Numerical values of CFU for *Figure 1—figure supplement 3*.

**Figure supplement 4.** Combined data for change in body weight during monotherapy and combination therapy in C3HeB/Fej mice .

**Figure supplement 4—source data 1.** Numerical values for body weight.

**Figure supplement 5.** C3HeB/FeJ Study 1 change in body weight during combination therapy in C3HeB/Fej mice, weight data.

**Figure supplement 5—source data 1.** Numerical values of body weight.

**Figure supplement 6.** C3HeB/FeJ Study 2 change in body weight during combination therapy in C3HeB/Fej mice, weight data.

**Figure supplement 6—source data 1.** Numerical values of body weight.

**Figure supplement 7.** C3HeB/FeJ Study 3 change in body weight during combination therapy in C3HeB/Fej mice, weight data.

**Figure supplement 7—source data 1.** Numerical values of body weight.

**Figure supplement 8.** BALB/c Study 1, CFU data.

**Figure supplement 8—source data 1.** Numerical values for CFU in Balb/c mice.

**Figure supplement 9.** BALB/c Study 2, CFU data.

**Figure supplement 9—source data 1.** Numerical values for CFU in Balb/c mice.

**Figure supplement 10.** Combined data for change in body weight during monotherapy and combination therapy in Balb/c mice.

**Figure supplement 10—source data 1.** Numerical values for body weight in Balb/c mice.

**Figure supplement 11.** BALB/c Study 1 change in body weight during combination therapy in Balb/c mice, weight data.

**Figure supplement 11—source data 1.** Numerical values for CFU in Balb/c mice.

**Figure supplement 12.** BALB/c Study 2 change in body weight during combination therapy in Balb/c mice, weight data.

**Figure supplement 12—source data 1.** Numerical values for CFU in Balb/c mice.

log$_{10}$) control, mice in the BPa (5.49±0.42 log$_{10}$), BPaL (4.76±0.38 log$_{10}$) and BPaS (4.98±0.35 log$_{10}$) treatment groups had significantly reduced the lung bacterial burden by 1.99, 2.72, and 2.50 log$_{10}$, respectively (*Figure 1E*). Although a higher CFU reduction was observed in the lungs of C3HeB/FeJ mice treated with BPaL or BPaS, these differences failed to achieve statistical significance compared to the BPa backbone regimen. All three regimens proved highly effective at reducing spleen bacterial burden in C3HeB/FeJ mice, with most mice returning no CFU within the limit of detection (LOD) employed herein (*Figure 1G*).

The effect of BPaL and BPaS (S=50 mg/Kg) combination therapy on the bacterial burden in the lungs and spleen of Mtb-infected BALB/c mice was determined at the end of 4 weeks of treatment. *Figure 1F, H* shows the combined lung and spleen CFU data from two independent studies (data for each study is shown in *Figure 1—figure supplements 8–9*). The combined result demonstrated that compared to UnRx (5.24±0.17 log$_{10}$) control, mice in the BPaL (1.29±0.13 log$_{10}$) and BPaS (1.44±0.17 log$_{10}$) treatment groups returned significantly fewer CFU in the lungs, with most mice returning no CFU within the LOD employed herein (*Figure 1F*). As in the C3HeB/FeJ TB model, no significant difference was observed in the lung CFU of BALB/c mice treated with either the BPaL or BPaS regimen. BPaL and BPaS therapy reduced BALB/c spleen bacterial burden to below the LOD of the assay with no CFU recovered for any treated mice (*Figure 1H*). In summary, these results support our hypothesis and demonstrate that both BPaL, BPaS (and BPa in the C3HeB/FeJ TB model) multidrug regimens show equivalent bactericidal effects in C3HeB/FeJ and BALB/c chronic TB efficacy models.

## Monitoring of adverse events

Five approaches were employed to monitor treatment-associated AEs in mice in this study including (1) changes in the body weight of mice; (2) lung histopathology and lesion scoring; (3) evaluation of complete blood count (CBC); (4) clinical pathology to study myelosuppression in the bone marrow and (5) changes in the content of immune cells in the lungs, spleen, bone marrow and blood.

## BPaL therapy decreases the body weight of mice

No significant difference in the body weight among the treatment groups in either C3HeB/FeJ or BALB/c mice was observed following 4 weeks of treatment with S or L alone (*Figure 1—figure supplements 4–7* and *Figure 1—figure supplements 10–12*, respectively). On the other hand, C3HeB/FeJ or BALB/c mice treated with BPa, BPaL, or BPaS showed marginal loss of body weight, ranging from 2.37–5.13% and only mice receiving the BPaL regimen, when compared to the UnRx control, reached statistically significant loses in body weight by the end of treatment (*Figure 1—figure supplements 4 and 10*, respectively).

## BPaL and BPaS therapy result in a significant lower lung lesion burden

The lesions in Mtb-infected UnRx C3HeB/FeJ mice (*Figure 2A and B*) showed a spectrum of diverse granuloma types ranging from aggregations of macrophages and lymphocytes to highly organized granulomas with collagen encapsulation and a region of central caseous necrosis that resembles to those found in some human patients (*Irwin et al., 2015*; *Driver et al., 2012*). By comparison, lung lesions of Mtb-infected UnRx BALB/c mice (*Figure 2C and D*) consisted of granulomas with a very homogeneous structure (*Irwin et al., 2015*) formed also by aggregations of macrophages and lymphocytes without a necrotic core. Mice treated for 4 weeks with the BPaL or BPaS regimen presented with a significant reduction in the number and size of granulomas in both C3HeB/FeJ and BALB/c TB models compared to their respective UnRx control, with no significant difference in lesion burden score between the combination drug treatment groups (*Figure 2E and F*).

## Association of L with altered blood profile and mild anemia in mice

The effect of L in the blood profile of humans and mouse has been reported (*Tang et al., 2015*; *Gerson et al., 2002*; *Mase et al., 2022*; *Bigelow et al., 2020*; *Bigelow et al., 2021*) but the same has not been reported for S. Therefore, a CBC profile was performed on Mtb-infected C3HeB/FeJ mice at the end of 4 weeks of treatment to quantify treatment-associated hematological changes. The results obtained from mice treated with L or S alone are summarized in *Figure 3A*. Of the 20-blood parameters evaluated, two blood parameters were affected during treatment. When compared to UnRx control and S-treated mice, L treatment significantly increased the red blood cell distribution width

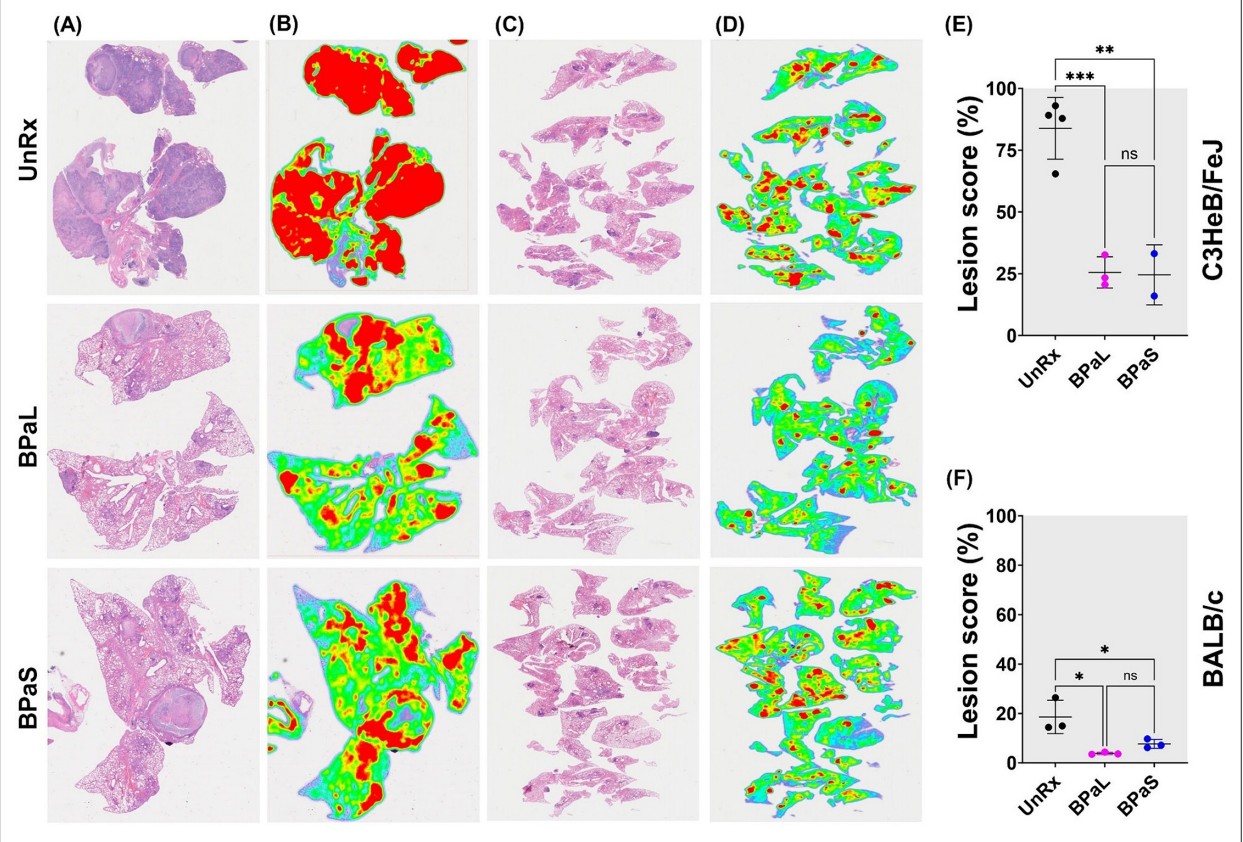

**Figure 2.** Effect of therapy on lung histopathology of TB mouse models after 4-weeks of treatment. At the end of therapy, the mice were euthanized, and their lungs were collected and processed for histopathology and lesion scoring. FFPE sections were cut at 5 μm, stained with hematoxylin and eosin (H and E) and imaged at 40x (**A**: C3HeB/FeJ, **C**: BALB/c). Lesion maps (**B**: C3HeB/FeJ, **D**: BALB/c) show the infected areas in red color while green color represents the uninvolved parenchymal tissue. Lesion scores (**E**: C3HeB/FeJ, **F**: BALB/c) were calculated as the proportion of infected area over the total lung area per animal. Statistical significance was calculated using one-way ANOVA with Tukey's multiple comparison test, and p < 0.05 was considered significant and ** = p<0.001, *** P<0.0001. UnRx = untreated.

The online version of this article includes the following source data for figure 2:

**Source data 1.** Lesion score for C3HeB/FeJ mice.

**Source data 2.** Lesion score for BALB/c mice.

standard deviation (RDWs), while both L and S treatment were associated with a significant decrease in the mean corpuscular hemoglobin concentration (MCHC) compared to UnRx control (*Figure 3A*).

The Nix-TB trial associated the long-term administration of L within the BPaL regimen as the causative agent resulting in anemia in patients treated with the BPaL regimen (*Conradie et al., 2020*). Thus, the effect of combination therapy with the BPaL or BPaS regimen on CBC profile was analyzed at 1-, 2-, and 4 weeks of treatment (*Figure 3B*). None of the 20 parameters of CBC changed during the first 2 weeks of treatment. However, out of the 20 blood parameters evaluated, a total of four blood parameters were affected at 4 weeks of treatment. L-containing BPaL regimen was again associated with a significant increase in the RDWs, and lower hemoglobin (HGB) compared to UnRx control after 4 weeks. This effect was not observed in mice treated with BPa or BPaS (*Figure 3B*). However, as in monotherapy, there was a trend towards lower overall MCHC in mice treated with either BPaL or BPaS. The mean platelet volume (MPV) was marginally higher at 4 weeks in mice treated with BPa compared to BPaL or BPaS, albeit not significantly different from UnRx control (*Figure 3B*). Given that no difference in HGB or RDWs was observed between UnRx control and their comparator BPa and BPaS regimens, we concluded that the significant HGB decrease and RDWs increase (often observed during the development of anemia) were associated with inclusion of L in the BPaL regimen.

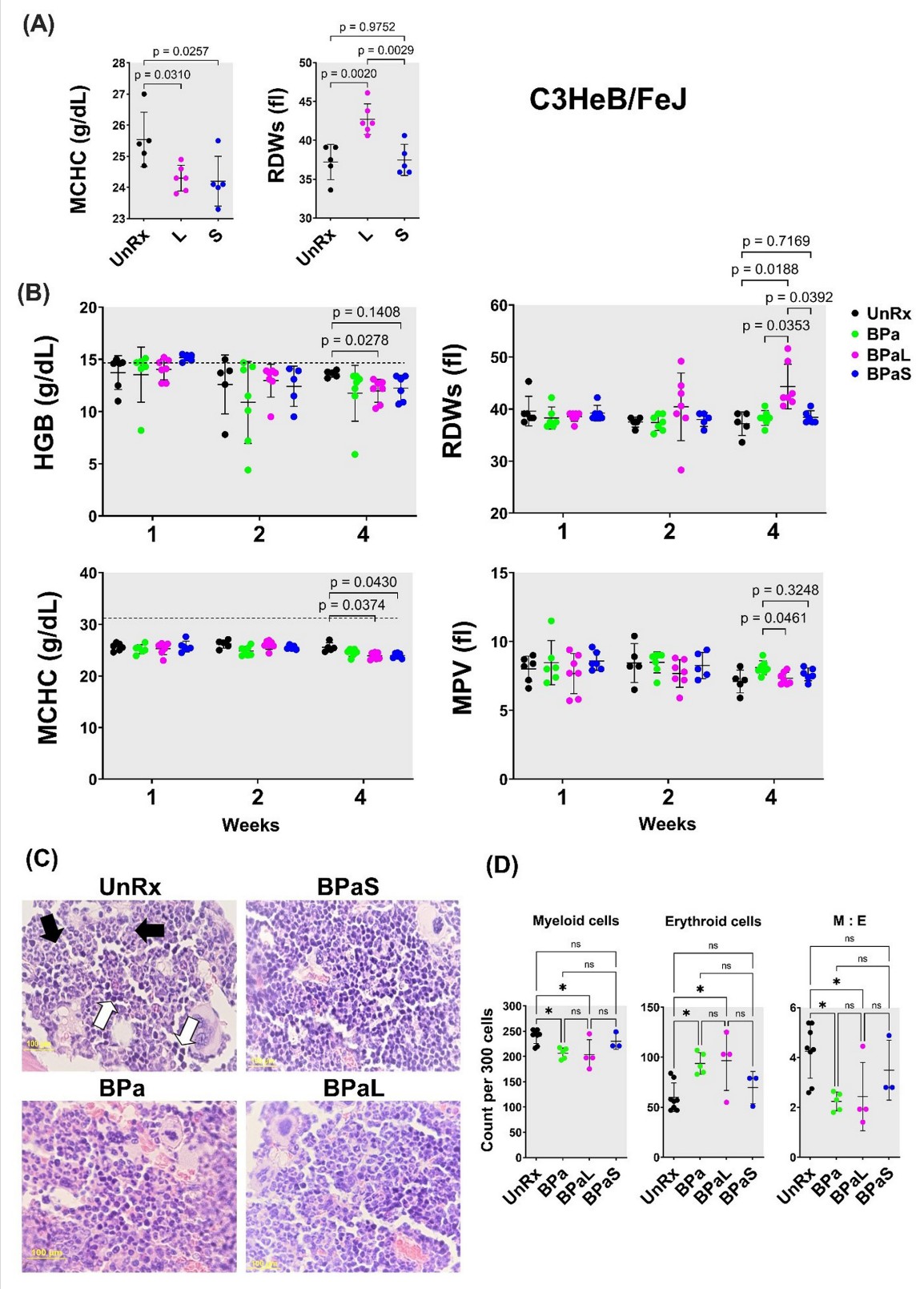

**(A)**

**C3HeB/FeJ**

**(B)**

UnRx
BPa
BPaL
BPaS

**(C)**

UnRx     BPaS

BPa     BPaL

**(D)**

Myeloid cells     Erythroid cells     M : E

**Figure 3.** Complete blood count profiling and bone marrow histopathology in C3HeB/FeJ TB mouse model during 4-weeks of therapy. During therapy of mice in *Figure 1*, the blood was collected at 1-, 2- and 4-weeks of treatment. The complete blood count was collected in VETSCAN HM5 hematology analyzer (Zoetis). (**A**) Monotherapy, (**B**) multidrug therapy. The MCHC (mean corpuscular hemoglobin concentration) and RDWs (red blood cell distribution width-standard deviation) along with the HGB (hemoglobin concentration) and MPV (mean platelet volume) are shown. A horizontal

*Figure 3 continued on next page*

*Figure 3 continued*

dotted line indicates the lower end of the reference interval for C3HeB/FeJ mice. The sternum, femur and tibias bones from each mouse were collected, fixed in 4% PFA, and processed for histology (**C** and **D**). Sections were cut at 5 μm, stained with hematoxylin and eosin (**H** and **E**) and imaged at 40x. (**C**) Representative photos of bone marrow sections showing myeloid (black arrows) and erythroid (white arrows) cells in bone marrow of untreated (UnRx) and treatment (BPa, BPaS and BPaS) groups. (**D**) The number of myeloid (M) and erythroid (E) among a total of 300 cells in 5 different regions were counted for each group and M:E was calculated. Statistical significance was calculated using one-way ANOVA with Tukey's test for multiple comparisons. p < 0.05 was considered significant and ** = p<0.001, *** = P<0.0001.

The online version of this article includes the following source data for figure 3:

**Source data 1.** Complete blood count for C3HeB/FeJ data.

**Source data 2.** Bone marrow histopathology in C3HeB/FeJ TB mice.

## Spectinamide 1599 recovers the altered ratio of myeloid to erythroid cells in bone marrow

To further evaluate if L was associated with myelosuppressive effect, we performed hematopathology analysis on bone marrow from Mtb-infected C3HeB/FeJ and BALB/c mice at the end of treatment. For C3HeB/FeJ mice, the number of myeloid (M) and erythroid (E) precursor cells were calculated from H&E stained sections and their myeloid to erythroid ratio (M:E) was determined by counting 300 bone marrow cells in five different regions (*Figure 3C*). The BPa and BPaL treatment significantly decreased myeloid cells while increasing the proportion of erythroid cells (*Figure 3D*). Hence, the corresponding ratio in the bone marrow of animals treated with BPaL or BPa was lower compared to the UnRx control. Importantly, the BPaS treatment did not show any difference in the content of myeloid or erythroid cells when compared to UnRx control suggesting that S in the BPaS was able to recover this effect. In Mtb-infected BALB/c mice, the number of myeloid and other cell types were counted, however, no significant difference was found among the control and treatment groups (data not shown).

## BPaL therapy increases proinflammatory cytokine response in bone marrow

A comparative analysis for the concentration of cytokines and chemokines in the bone marrow, plasma, and lung samples from Mtb-infected C3HeB/FeJ mice treated with BPaL or BPaS regimen was also conducted. The bone marrow samples demonstrated a significant difference between the BPaL and BPaS groups, with appreciably higher level of pro-inflammatory cytokines and chemokines (IL-1β, IL-12p70, IL-23, TNFα, GROα (CXCL1), MP-2α (CXCL2), IP-10 (CXCL10), MP-1α (CCL3), RANTES (CCL5), MCP-3 (CCL7), and Eotaxin (CCL11)) in the BPaL- compared to BPaS-treated mice (*Figure 4B*). The plasma and lung samples, however, had similar cytokine and chemokine contents the treatment groups except for MCP-3 (CCL7) in plasma which was significantly higher in BPaS compared to the BPaL group (*Figure 4—figure supplement 1*). We also performed a correlation analysis of bone marrow cytokine and chemokine content with lung CFU obtained from treatment (BPaL or BPaS) and UnRx groups (*Figure 4C*). The analysis suggested that compared to UnRx control, there was a strong correlation between the profound reduction of lung bacterial burden and the profound reduction in bone marrow cytokine and chemokine contents observed in mice from BPaL or BPaS groups. Similar correlations were found between lung CFU and content of cytokines and chemokines in lung (*Figure 4—figure supplement 2*) and plasma (*Figure 4—figure supplement 3*).

## BPaL and BPaS therapies reduce inflammation-associated cells

We further assessed the environment of immune cells using flow cytometry in bone marrow, lungs, and blood from each group of Mtb-infected C3HeB/FeJ mice (*Figure 5*). The bone marrow (*Figure 5A*) results revealed that as compared to UnRx control, there was a significant decrease in the percentage of inflammatory myeloid phenotypes (CD45 +CD3-CD11b+CD11c-Ly6C+CCR2+, CD45 +CD3-CD11b+CD11c-Ly6C+CCR2+MHC-II+and CD45+CD3-D14+CCR2+) in response to therapy with either BPa, BPaL, or BPaS. In contrast, neutrophils (CD45 +CD3-CD11b+CD11c-Ly6C+Ly6G[high]), precursor T cells (CD45 +CD3+), and B cells (CD45 +CD3 CD19+B220-) were significantly increased in either BPa, BPaL, or BPaS treatment groups compared to UnRx control in bone marrow. This reduced inflammatory response in treatment groups is also consistent in blood shown by significantly

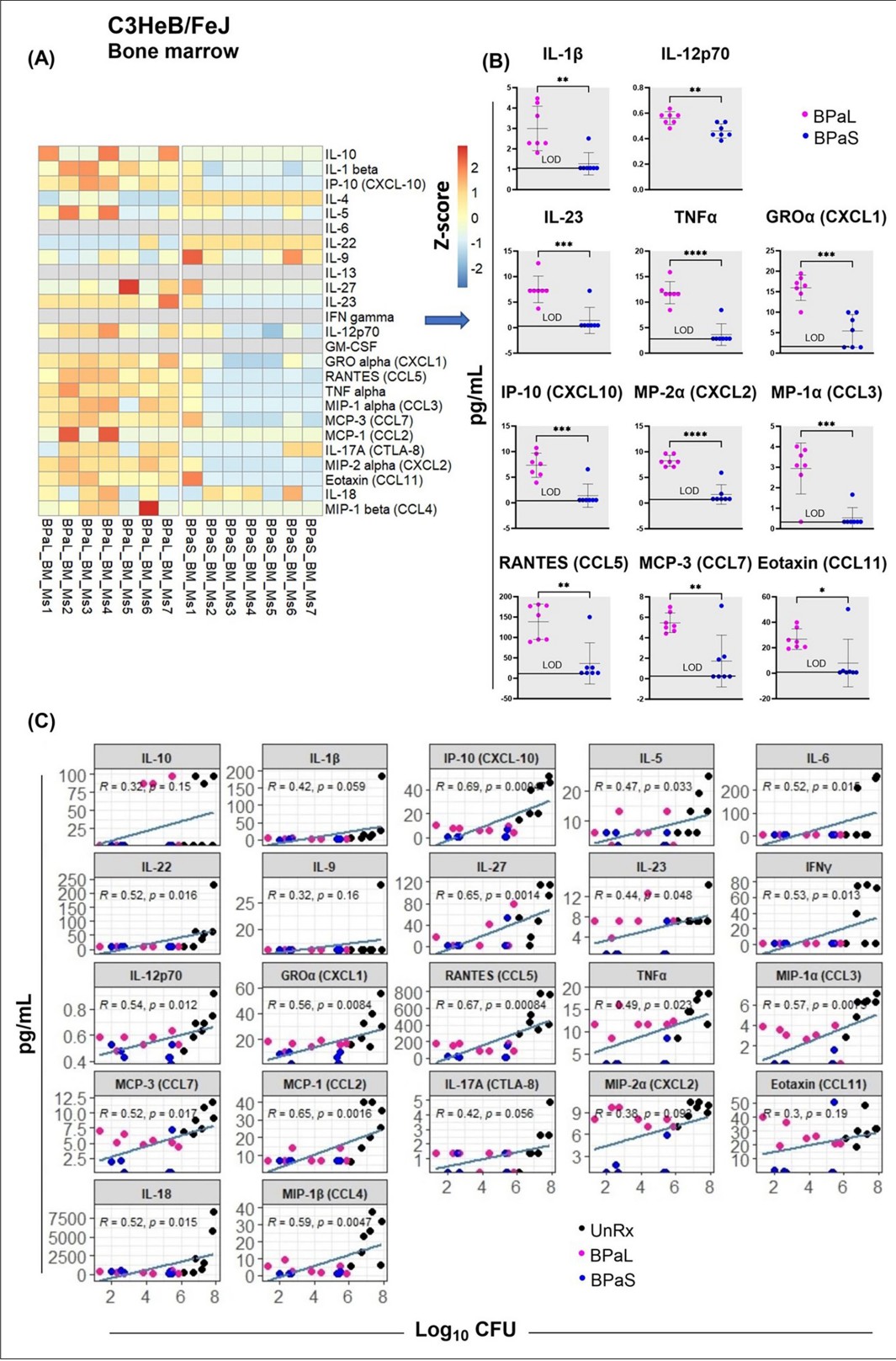

**Figure 4.** Bone marrow cytokine and chemokine profiling in C3HeB/FeJ TB mouse model after 4-weeks of treatment. Femur bones from selected studies in mice in *Figure 1* were collected to harvest the bone marrow. The bone marrow was resuspended in PBS, centrifuged and the supernatant was collected for the evaluation of cytokine's content. BPaL and BPaS therapy showed profile of 26 cytokines and chemokines in bone marrow and the

*Figure 4 continued on next page*

*Figure 4 continued*

data were converted to Z score and represented as a heatmap (**A**) and graphically (**B**). (**C**) Spearman's correlation analysis of bone marrow cytokines and chemokines (Y axis; pg/ml) with the lung bacterial burden (X axis; $log_{10}$CFU). Statistical significance was calculated using the t test. p < 0.05 was considered significant and ** = p<0.001*** = p<0.0001, **** = P<0.00001.

The online version of this article includes the following source data and figure supplement(s) for figure 4:

**Source data 1.** Numerical values of bone marrow cytokine and chemokine profiling in C3HeB/FeJ TB mouse model after 4-weeks of treatment.

**Figure supplement 1.** Change in the cytokines and chemokines profile in *Mycobacterium tuberculosis* infected C3HeB/FeJ mice during drug treatment.

**Figure supplement 2.** Spearman's correlation analysis between bacterial burden colony-forming units (CFU) and cytokine and chemokine profile in the lungs of *Mycobacterium tuberculosis* infected C3HeB/FeJ mice treated with BPaL and BPaS regimen.

**Figure supplement 3.** Spearman's correlation analysis between lung bacterial burden colony-forming units (CFU) and plasma cytokine and chemokine profile in *Mycobacterium tuberculosis* infected C3HeB/FeJ mice treated with BPaL and BPaS regimen.

---

reduced inflammatory myeloid cells (CD45 +CD3-CD11b+CD11c-Ly6C+CCR2+) (*Figure 5B*). Interestingly, the response to therapy in the lungs (*Figure 5C*) was manifested by a significant increase of CD3 +CD4+T helper cells and B-1 cells (CD3-CD19+) and a reverse trend for CD3 +CD8+and γδ-T cells (CD3 +CD8+γδTCR+).

Furthermore, we also assessed changes in the distribution of immune cells in the lungs from Mtb-infected C3HeB/FeJ mice in response to therapy using multiplex fluorescent immunohistochemistry (mfIHC). A seven-color composite image for cell markers (B220, CD4, CD8, Foxp3, F4/80, and Ly6G) along with DAPI is shown in *Figure 6A*, while their single-color staining is shown in *Figure 6B*. *Figure 6A* shows a typical necrotic TB granuloma comprised of central necrosis, peripheral rim, and lung parenchyma. The analysis of mfIHC images revealed that the BPaL and BPaS treatments significantly and dramatically lowered the number of neutrophils (count based on Ly6G+) compared to UnRx control, however, F4/80+ cells were observed significantly higher in BPaS compared to UnRx control (*Figure 6—figure supplement 1*). Interestingly, the spearman's correlation plot (*Figure 6D*) shows that a significant decrease in neutrophils was positively correlated with the corresponding increase in most of the other immune cells.

## Discussion

We used two preclinical chronic TB murine efficacy models (*Harper et al., 2012*; *Irwin et al., 2014*; *Aly et al., 2006*) to investigate BPa, BPaL, and BPaS, for efficacy and any associated AEs during the course of 4 weeks of treatment. Overall, our antimicrobial data are in accordance with the granuloma spectrum of both mouse models used where a more robust reduction in lung bacterial burden was observed in the absence (BALB/c) versus the presence (C3HeB/FeJ) of necrotic lesions. Both multidrug regimens such as BPaL or BPaS significantly reduced lung bacterial burden by 3.8–4.0 $log_{10}$ and 2.5–2.7 $log_{10}$ in BALB/c and C3HeB/FeJ TB models, respectively (*Figure 1E–H*). We, therefore, conclude that both regimens promote similar bactericidal effects in murine models lacking, or featuring, advanced pulmonary pathology. Furthermore, the potent antimicrobial effect of the BPaL and BPaS regimens also resulted in improvement of the pathological outcome during chronic infection with Mtb but only the L-containing BPaL regimen, not BPaS, was associated with a significant decrease in the body weight at week 4 in Mtb-infected BALB/c and C3HeB/FeJ mice (*Figure 4*; *Figure 1—figure supplements 4 and 10*). Future studies will determine if prolonged treatment with BPaL will continue affecting the body weight of the animals. The extent of weight loss is an important preclinical and clinical parameter in TB patients because it determines the severity of disease progression (*van Crevel et al., 2002*), and it is also an indicator of *in vivo* drug efficacy (*Nikonenko et al., 2004*). The bactericidal effects of S (as monotherapy) (*Pstragowski et al., 2017*) and BPaL (*Nuermberger et al., 2022*) in mice observed in these studies are in line with previous reports and a similar inverse relationship between body weight and L exposure was recently reported in human patients (*Wasserman et al., 2019*).

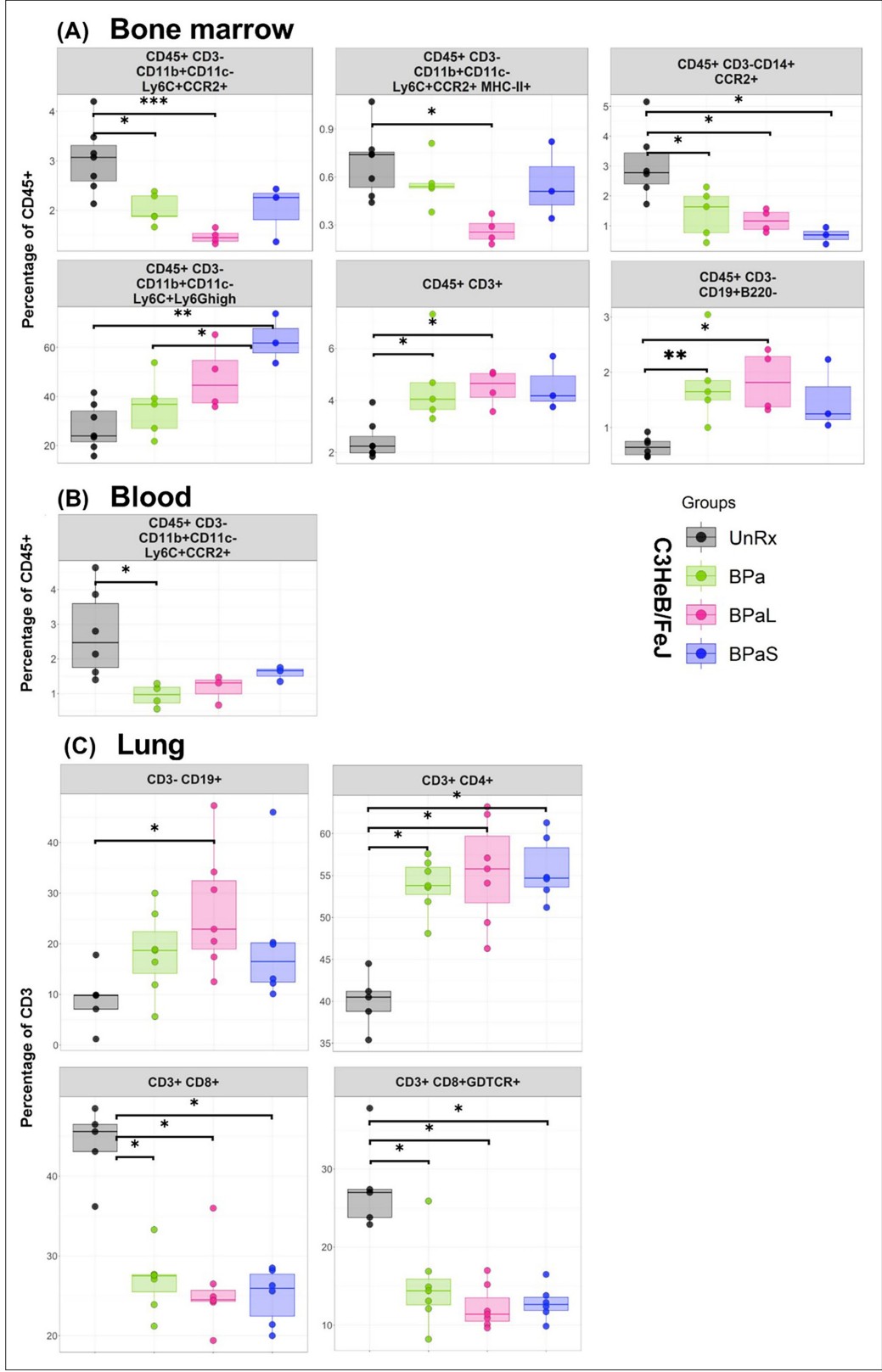

**Figure 5.** Immune cell populations in the bone marrow, lung and blood of C3HeB/FeJ TB mouse model after 4-weeks of treatment. The bone marrow, lung and blood from selected studies from *Figure 1* were evaluated by flow cytometry. The samples were processed for a panel of 17-color antibodies and the data were analyzed by FlowJo software using manual gating strategy. The myeloid and lymphoid phenotypes present in the untreated

*Figure 5 continued on next page*

*Figure 5 continued*

(UnRx) and treatment (BPa, BPaL or BPaS) groups are shown. Statistical significance was calculated using one-way ANOVA with Tukey's test for multiple comparisons. p < 0.05 was considered significant and ** = p<0.001*** = p<0.0001, **** = P<0.00001.

The online version of this article includes the following source data for figure 5:

**Source data 1.** Numerical values for immune cell populations in the bone marrow, lung and blood of C3HeB/FeJ TB mouse model after 4-weeks of treatment.

Among L-associated hematological side effects, the incidence of anemia is reported up to 62.5% in MDR and XDR-TB patients (*Dayyab et al., 2021*; *Palomino and Martin, 2014*; *Sotgiu et al., 2012*) and the onset of this effect can occur at 2 weeks to 2 months of L administration (*Tang et al., 2015*). In TB patients treated with anti-TB drugs (*Mirlohi et al., 2016*; *Kassa et al., 2016*; *Luo et al., 2022*), increased RDWs and lower HGB are associated with anemia and these parameters serve as markers of disease prognosis. Using a CBC profile of 20 peripheral blood parameters, our preclinical study failed to detect any difference between L or S 4- weeks monotherapy in terms of total red blood cells, white blood cells, platelets, or hemoglobin (HGB) concentration compared to UnRx control (data not shown). However, L treatment alone increased the RDWs and both L and S decreased the MCHC (*Figure 3A*). Likewise, none of the 20 blood parameters evaluated showed any change after the first 2 weeks of therapy with BPa, BPaL, or BPaS (*Figure 3B*). However, by 4 weeks, a significant drop in HGB and an increase in the RDWs was apparent for the BPaL group (*Figure 3B*). This is interpreted to mean that mild hematological effects observed in mice treated for 4 weeks with L or BPaL are dependent on the number of L-doses administered and are thus, time-dependent. Overall, the onset of these hematological effects when testing an L-containing regimen (BPaL) is in agreement with previous studies (*Tang et al., 2015*; *Gerson et al., 2002*) in human patients.

The mechanism of L-induced toxicity is attributed to its binding with the host mitochondrial ribosomes leading to mitochondrial toxicities (*De Vriese et al., 2006*). The latter results in the activation of Nlrp3 inflammasome (*Iyer et al., 2013*) and subsequently results in L-mediated bone marrow myelosuppression *Winchell et al., 2020*; a phenomenon consistent with the hematologic anomalies seen in patients treated with L for extended time periods. Because spectinamides do not bind to mitochondrial ribosomes there is reduced potential for similar side effects (*Lee et al., 2014*; *Modongo et al., 2015*). To conclusively test this hypothesis, we performed bone marrow histopathology to quantify the myeloid to erythroid ratio (M:E), a parameter that provides information about the relative proportions of myeloid lineage (granulocytes, monocytes, and their precursors) to erythroid lineage (*Elmore, 2006*). The BPa and BPaL regimens altered M:E in the C3HeB/FeJ TB model by suppressing myeloid and inducing erythroid lineages (*Figure 3C&D*) whereas no such difference was observed in mice treated with BPaS compared to untreated control. L was previously shown to impact M:E ratio, although an opposite trend was observed in those studies, which employed 12- days of L administration and a different strain of otherwise healthy mice (*Iyer et al., 2013*). Time course studies using a single consistent assay method are needed to resolve this discrepancy.

Elevation of interleukin 1β (IL-1β) levels and activation of Nlrp3 inflammasome have been previously linked to myelosuppression (*Winchell et al., 2020*). A 26-plexed immunoassay on bone marrow samples from Mtb-infected C3HeB/FeJ mice revealed that most of the proinflammatory cytokines and chemokines including IL-1β, IL-12p70, and TNF-α were present at significantly higher concentrations in animals treated with BPaL compared to BPaS (*Figure 4A and B*). The presence of elevated IL-1β was previously reported during monotherapy with L (*Iyer et al., 2013*; *Jasenosky et al., 2015*). Studies from UnRx Mtb-infected C3HeB/FeJ mice also revealed highly elevated levels of cytokines and chemokines (*Figure 4C*). As expected, the positive therapeutic effect of BPaL and BPaS (as seen by the reduction in bacterial and lesion burden of lungs) also correlated with decreased levels of proinflammatory cytokines in bone marrow, lung, and plasma (*Figure 4—figure supplements 2–3*, respectively).

Similar to changes observed for cytokine profile, the flow cytometry data of bone marrow, blood, and lungs along with quantification of mfIHC lung image analysis revealed a significant change in the percentage of myeloid and lymphoid phenotypes during treatment compared to UnRx control (*Figures 5 and 6*). Most notably, the therapeutic effect of the BPa, BPaL, and BPaS treatments reduced inflammatory myeloid cells expressing CCR2 in blood and bone marrow and reduction in cytotoxic

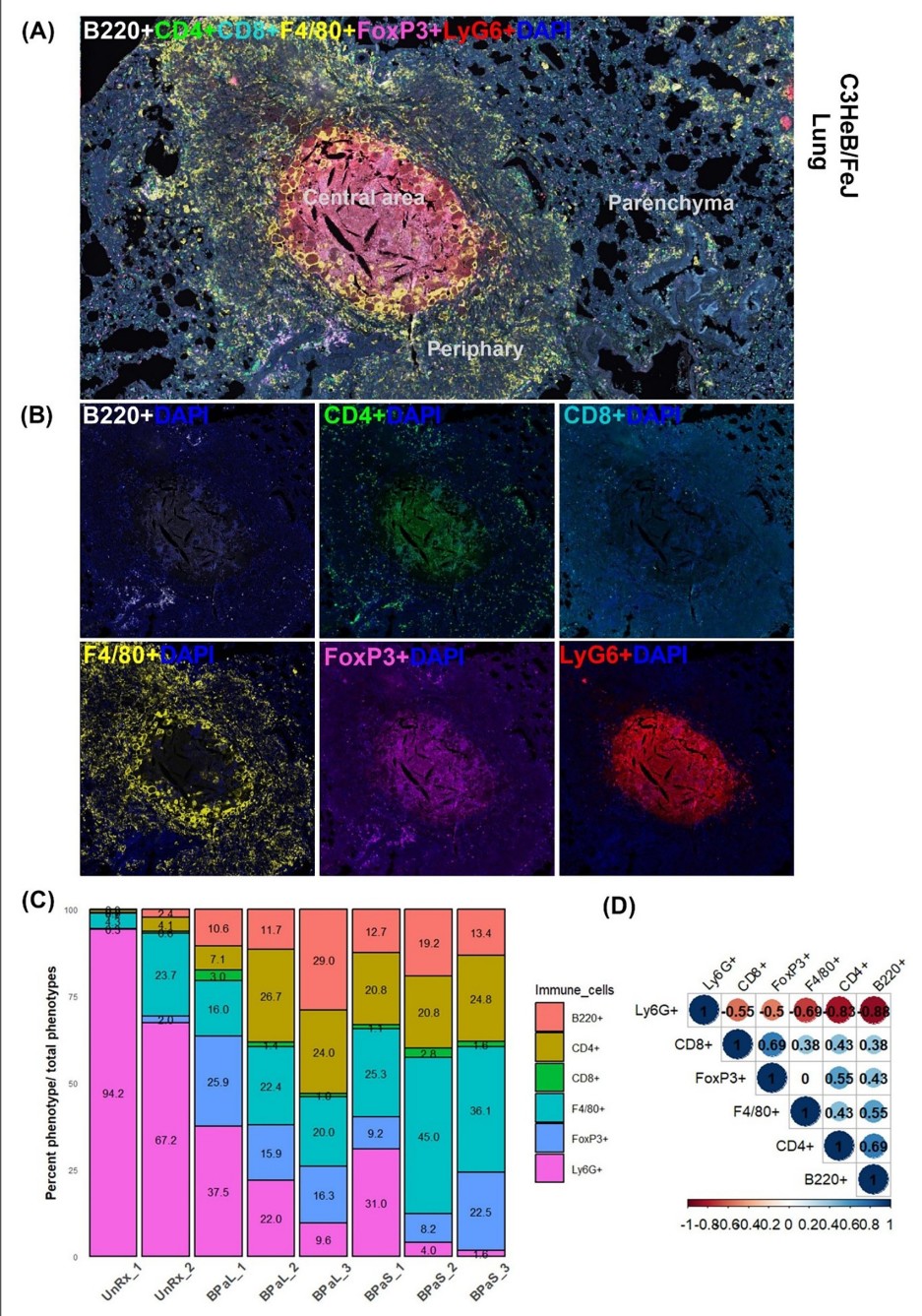

**Figure 6.** Immune cell populations in the lungs of C3HeB/FeJ TB mouse model after 4-weeks of treatment. Selected mice from those shown in *Figure 1* were processed for multiplex fluorescence immunohistochemistry (mfIHC). The mfIHC was performed for a panel of 6-color antibodies + DAPI using Opal-plex Tyramide Signal Amplification (TSA). Slides were scanned using multispectral automated PhenoImager (Akoya Biosciences) and analyzed for different immune cell populations using the inForm tissue Finder and Phenochart software (Akoya Biosciences). (**A**) The lung mfIHC full composite image displays B220, CD4, CD8, F4/80, FoxP3 and Ly6G markers along with DAPI staining for nuclei in the TB granuloma. The central and peripheral regions of a TB granuloma and the parenchyma of lung are also shown. (**B**) Single color composite image of individual markers with DAPI showing distribution of each immune cell population in the TB granuloma. (**C**) Cell populations (%) of several immune cells per total number of phenotypes calculated in untreated (UnRx: n = 2 mice) and treatment (BPaL and BPaS: n = 3 mice each) groups based on a panel of 6-color antibodies + DAPI. (**D**) Spearman's correlation matrix for several immune cell populations (B220, CD4, CD8, F4/80, FoxP3, Ly6G) showing all relationships. A coefficient with a value

*Figure 6 continued on next page*

*Figure 6 continued*

of either +1 (blue), 0 (white), or -1 (red) indicates a perfect association, no association, and a perfect negative association of ranks, respectively. Numbers indicate the correlation coefficient.

The online version of this article includes the following source data and figure supplement(s) for figure 6:

**Source data 1.** Numerical values for immune cell populations in the lungs of C3HeB/FeJ TB mouse model after 4-weeks of treatment.

**Figure supplement 1.** Immune cell populations in the lungs of *Mycobacterium tuberculosis* infected C3HeB/FeJ tuberculosis (TB) model after 4 weeks of therapy.

T cells (CD3 +CD8+) and γδT cells (CD3 +CD8+γδTCR+) in lungs. In addition, BPa, BPaL, and BPaS therapy significantly increased the influx of helper T cells (CD3 +CD4+), regulatory T cells (Foxp3+), and B cells (CD3-CD19+) in lungs. These results suggest that the combination therapy promotes the immune system's equilibrium by reducing inflammation and enhancing adaptive immune responses.

An additional striking finding from this study was a strong concordance between the decrease in lung and spleen bacterial burden and a corresponding decrease in the number of cells expressing the neutrophil-associated marker (Ly6G) (*Figure 6C and D*). An implication of this finding is that the favorable treatment outcomes may promote a corresponding decline in the Ly6G neutrophil population as suggested before (*Lovewell et al., 2021*). Among all immune cell phenotypes studied, no differences were found between BPaL and BPaS treated animals, and only the F4/80+ cells in the BPaS (but not in the BPaL) treated animals showed a significant increase when compared to the UnRx control.

To conclude, the TB drug development field is working towards developing shorter and safer therapies with a common goal of developing new multidrug regimens of low pill burden that are accessible to patients, of short duration (ideally 2–3 months), and consist of 3–4 drugs of novel mode-of-action with proven efficacy, safety, and limited toxicity. Here, we present initial results for new multidrug regimens containing inhaled spectinamide 1599 that are in line with these goals. It is proposed that the human use of spectinamides 1599 will be administered using a dry powder formulation delivered by the RS01 Plastiape dry powder inhaler. We already reported on the aerodynamic properties of dry powder spectinamide 1599 within #3 HPMC capsules and delivered from a RS01 Plastiape inhaler device (*Stewart et al., 2019*). Future studies to understand the pharmacokinetics of mono, binary, and ternary combinations of BPaS are underway. These studies also aim to identify the optimal dose level and dosing frequency of each regimen along with their efficacy and relapse-free-sterilization potential. Studies are also planned to use a model-based pharmacokinetic-pharmacodynamic (PKPD) framework, guided by an existing human BPa PKPD model (*Lyons, 2022*; *Lyons et al., 2024*) to find allometric human dose levels, dosing frequencies, and treatment durations that will inform the experimental design of future clinical studies.

## Materials and methods

Female C3HeB/FeJ and BALB/c mice at 6–8 weeks of age were purchased from the Jackson Laboratories. All protocols (PARF 16-047B) and the use of these animals were approved by the Institutional Animal Care and Use Committee (IACUC; 1508) at CSU. Animals were infected with a low-dose aerosol infection of Mtb (Erdman strain; ATCC 35801) using an inhalation exposure system (Glas-Col, Terre Haute, IN) calibrated to deliver ~50–100 colony-forming units (CFU) to the lungs (*Gonzalez-Juarrero et al., 2021*). Clinical observations (e.g. inactivity, rough fur, hunched posture, increased respiratory rate or effort) were monitored daily and their body weights were taken weekly.

### Drug preparation and treatment

Bedaquiline fumarate (B) and linezolid (L) were obtained from LKT laboratories, pretomanid (Pa) from ChemShuttle, and 1599 (S: dihydrochloride) was provided by Dr. Lee at St. Jude Children's Research Hospital. B was administered at 25 mg/kg. Pa and L were administered at 100 mg/kg each and S was delivered by inhalation in liquid form at 50 and 100 mg/kg to BALB/c and C3HeB/FeJ mice, respectively. The drugs were formulated in weekly batches according to the body weight of the animals, aliquoted for single daily dosing, and stored at 4 °C in the dark.

The drugs were prepared and administered as reported previously (*Robertson et al., 2017*; *Gonzalez-Juarrero et al., 2021*). Drug treatment was started 4 weeks post-aerosol infection for

**Table 1.** Itemized list of antibodies for flow cytometry.

| Reagent or resource | Source | Identifier |
|---|---|---|
| Antibodies | | |
| Anti-mouse LY6G PerCP | BioLegend | Cat# 127654; RRID: AB_11218876 |
| Anti-mouse CD14 PerCP Cy5.5 | Invitrogen | Cat# 120606; RRID: AB_493267 |
| Anti-mouse NKp46/CD335 PE | BioLegend | Cat# 137604; RRID: AB_2566163 |
| Anti-mouse B220/CD45 R PE-Cy7 | BioLegend | Cat# 103222; RRID: AB_2573837 |
| Anti-mouse CD8 FITC | BioLegend | Cat# 100706; RRID: AB_394458 |
| Anti-mouse CD34 PE-Dazzle 594 | BioLegend | Cat# 128616; RRID: AB_11219403 |
| Anti-mouse TER119 APC | BD Pharmingen | Cat# 561033; RRID: AB_10900980 |
| Anti-mouse γδ-TCR APC Fire 750 | BioLegend | Cat# 118129; RRID: AB_755986 |
| Anti-mouse LY6C Alexa Fluor 700 | BioLegend | Cat# 128024; RRID: AB_2869739 |
| Anti-mouse CD4 BV421 | BioLegend | Cat# 100544; RRID: AB_2562555 |
| Anti-mouse MHC-II BV480 | BD Biosciences | Cat# 566088; RRID: AB_2562612 |
| Anti-mouse CD11b Pacific Blue | BioLegend | Cat# 101224; RRID: AB_2565937 |
| Anti-mouse CD3e BV510 | BioLegend | Cat# 100353; RRID: AB_2563056 |
| Anti-mouse CD45 BV570 | BioLegend | Cat# 103136; RRID: AB_2814047 |
| Anti-mouse CD19 BV605 | BioLegend | Cat# 115540; RRID: AB_2563289 |
| Anti-mouse CCR2 BV711 | BD Biosciences | Cat# 747964; RRID: AB_2660295 |
| Anti-mouse CC11c BV785 | BioLegend | Cat# 117335; RRID: AB_2073247 |

BALB/c and at 8–9 weeks post-aerosol infection for C3HeB/FeJ mice to allow time for lung pathology to fully develop. All drugs were administered once daily for 5 days/week for 4 weeks by oral (gavage) administration except S which was administered 3 days/week by intrapulmonary aerosol delivery using the Penn Century microsprayer as reported previously (*Gonzalez-Juarrero et al., 2021*). B was administered in the morning, Pa 1 hr after B and, L and S at least 4 hr after Pa.

## Necropsy

After 4-weeks of treatment, C3HeB/FeJ and BALB/c mice were humanely euthanized by $CO_2$ narcosis. Blood, lungs, spleen, femur, and tibia bones were collected from each mouse for further processing and analysis.

## Assessment of efficacy

The efficacy of the treatment was assessed by determining changes in bacterial burden [measured as CFU] in the lungs and spleen of animals at necropsy. The lungs and spleen were homogenized and prepared as reported previously (*Gonzalez-Juarrero et al., 2021*). The lung homogenates were plated onto 7H11 agar plates supplemented with 0.4% activated charcoal to reduce the carryover effect of drugs and incubated at 37 °C for 6–8 weeks before the final CFU count. The remaining lung homogenate was centrifuged, and the supernatant was collected and stored at –80 °C for evaluation of cytokines and chemokines.

## Histopathology and lesion scoring

The lungs were fixed in 4% paraformaldehyde (PFA) for 48 hr and then embedded in paraffin for histopathology. Sections from formalin-fixed and paraffin-embedded (FFPE) tissues were cut at 5 µm, stained with hematoxylin and eosin (H&E), and scanned at 40 X magnification using multispectral automated PhenoImager (Akoya Biosciences) for histopathological evaluation. The extent of lung lesion burden was quantified in blinded digital images using an open-source QuPath software for image analysis as described previously (*Dutt et al., 2022*). For each tissue section, a region of interest

(ROI) was generated at low magnification with a custom tissue-detecting algorithm using decision forest training and classification to differentiate tissue versus background based on color and area. Lesions were identified within tissue ROIs at high magnification with an additional custom-made algorithm using decision forest training and classification based on staining intensity, color normalization and deconvolution, area, and morphological features. Percent lesion calculations were integrated into the same algorithm and calculated from tissue area and lesion area as designated by the ROI and lesions detected.

The sternum and one femur were fixed in 4% PFA and processed for histology. To evaluate the myelosuppressive effect of the drugs, bone sections were cut at 5 µm and stained with H&E. The number of myeloid and erythroid cells from 5 different regions of the bone were blinded and then counted by a veterinary clinical pathologist.

## Bone marrow collection

Briefly, a 0.6 mL sterile Eppendorf tube punctured at the bottom with the help of a 26-gauge needle was inserted into a 1.5 mL sterile Eppendorf tube. One end of the epiphysis of the long bones was cut open to expose the bone marrow and placed down into the small Eppendorf tube system. The tubes were centrifuged at 10,000 x g for 15 s and the marrow was collected from the base of the large Eppendorf tube. The bone marrow was resuspended in PBS and centrifuged again. Thereafter, the supernatant was collected and stored at –80 °C for evaluation of cytokines and chemokines while the bone marrow cells were saved in 4% PFA and freezing media for further use in clinical pathology analysis and flow cytometry, respectively.

## Processing of blood

For CBC analysis of C3HeB/FeJ animals during the treatment, blood was collected in K2-EDTA tubes via submandibular vein puncture as described previously (*Golde et al., 2005*). The blood was immediately analyzed in a VETSCAN HM5 hematology analyzer (Zoetis).

At the time of necropsy, whole blood was collected via cardiac puncture in K2-EDTA-containing tubes. After adding an equal volume of PBS, the samples were centrifuged at 800×g for 10 min at 25 °C with the brake off (deceleration = 0). The top plasma layer was collected and stored at –80°C for evaluation of their cytokine's content. The buffy coat was collected, and washed and the erythrocytes were lysed using Miltenyi RBC lysis buffer (Miltenyi, CA). The cells were washed and resuspended in 500 µL of complete DMEM media and prepared for flow cytometry analysis.

## Cytokine quantification

Multiplex immunoassay was performed using a Luminex bead-based multiplex ELISA kit (ProcartaPlex Mouse Cytokine & Chemokine Panel 1 26plex, reference # EPXR260-26088-901, Invitrogen). Each sample was normalized to the total protein concentration determined by Bicinchoninic acid (BCA) assay (Thermo Fisher). The BCA and Luminex assay were performed according to the manufacturer's instructions and the final stained samples were fixed with 4% PFA prior to acquisition. Sample data were acquired on a MAGPIX instrument running xPONENT 4.3 software (Luminex Corp.). Heatmaps were generated using the R pheatmap package. Correlation analysis of cytokine contents in bone marrow, plasma, and lungs with the lung bacterial burden was performed using the corrplot package in R.

## Immune cell population analysis

Single-cell suspension of bone marrow, blood, and lungs from C3HeB/FeJ mice was prepared as described previously (*Dutt et al., 2022*). Cells counting, viability staining, and cell staining (*Table 1*) was performed accordingly (*Dutt et al., 2022*). Samples were acquired using Cytek Aurora 4-Laser spectral flow cytometer where 100,000 events were recorded. Data were analyzed in FlowJo software (BD Biosciences) using manual gating (*Fox et al., 2020*).

## Multiplex fluorescence immunohistochemistry

Five µm sections of FFPE lung tissues were stained for multiplex fluorescence immunohistochemistry (mfIHC) by the Imaging Core at the University of Colorado, Anschutz Medical Campus, Denver. The mfIHC was performed for a panel of 6-color antibodies + DAPI using the Opal-plex Tyramide Signal

Amplification (TSA) technique using a Leica Bond III autostainer. The details of antibodies and Opal fluorophores used are given in *Supplementary file 1*. Each antibody was optimized using Opal 3-Plex Anti-Rb Detection Kit (Akoya Biosciences Inc cat# NEL830001KT) and stained with automated LabSat Research (Lunaphore Technologies SA, Epredia). Slides were scanned using multispectral automated PhenoImager (Akoya Biosciences) and analyzed for several immune cell populations using the inForm tissue Finder (Version 2.4.8) and Phenochart (Version 1.0.12) software (Akoya Biosciences).

## Statistical analysis

Bacterial burden data were expressed as CFU which were $Log_{10}$-transformed and analyzed using GraphPad Prism version 9.5.1 (GraphPad Software, La Jolla, CA). The statistical analysis was performed using a Tukey's multiple comparison test as part of either one-way or two-way ANOVA and mixed-model effect where necessary. The correlation analysis was performed using the spearman's correlation test. Flow cytometry and mfIHC data were graphed in R studio and statistical evaluation was performed using stats package in R.

## Acknowledgements

This research was supported by the National Institute of Allergy and Infectious Diseases and the Office of the Director of the National Institutes of Health (grants R01AI120670, R01AI090810, and 1S10OD030263, respectively). Dr. Malik Zohaib Ali was a Fulbright Foreign Student Program grantee from Pakistan. Dr. Charles Daley from the Division of Mycobacterial and Respiratory Infections, National Jewish Health, Denver, Colorado 80206, USA advised on current TB drug treatments. The content is solely the responsibility of the authors and does not necessarily represent the official views of the National Institutes of Health. We acknowledge the staff of the Laboratory Animal Resources at Colorado State University for providing animal care.

## Additional information

### Funding

| Funder | Grant reference number | Author |
| --- | --- | --- |
| National Institute of Allergy and Infectious Diseases | R01AI120670 | Anthony J Hickey Bernd Meibohm Mercedes Gonzalez Juarrero |
| National Institute of Allergy and Infectious Diseases | R01AI090810 | Richard Lee |
| Office of Research Infrastructure Programs | 1S10OD030263 | Mercedes Gonzalez Juarrero |
| Fulbright U.S. Student Program | Pakistan | Malik Zohaib Ali |

The funders had no role in study design, data collection and interpretation, or the decision to submit the work for publication.

### Author contributions

Malik Zohaib Ali, Conceptualization, Data curation, Software, Formal analysis, Validation, Investigation, Visualization, Methodology, Writing – original draft; Taru S Dutt, Data curation, Software, Formal analysis, Methodology, Writing – original draft; Amy MacNeill, Conceptualization, Data curation, Formal analysis, Supervision, Validation, Investigation, Methodology, Writing – original draft; Amanda Walz, Formal analysis, Methodology; Camron Pearce, Software, Formal analysis, Investigation, Methodology; Ha Lam, Software, Investigation, Methodology; Jamie S Philp, Formal analysis, Investigation, Methodology; Johnathan Patterson, Software, Formal analysis, Methodology; Marcela Henao-Tamayo, Resources, Data curation, Supervision; Richard Lee, Conceptualization, Supervision, Funding acquisition, Writing – review and editing; Jiuyu Liu, Data curation, Software, Validation, Investigation, Methodology; Gregory T Robertson, Conceptualization, Validation, Writing – original draft, Writing

– review and editing; Anthony J Hickey, Conceptualization, Resources, Software, Supervision, Funding acquisition, Writing – review and editing; Bernd Meibohm, Conceptualization, Resources, Funding acquisition, Validation, Project administration, Writing – review and editing; Mercedes Gonzalez Juarrero, Conceptualization, Resources, Data curation, Formal analysis, Supervision, Funding acquisition, Validation, Investigation, Visualization, Methodology, Writing – original draft, Project administration, Writing – review and editing

## Author ORCIDs
Malik Zohaib Ali (iD) http://orcid.org/0009-0004-1889-5970
Mercedes Gonzalez Juarrero (iD) https://orcid.org/0000-0002-4045-2365

## Ethics
All protocols (#16-047B) and use of these animals (#1508) were approved by the Institutional Animal Care and Use Committee (IACUC) at CSU.

Reviewer #1 (Public Review): https://doi.org/10.7554/eLife.96190.3.sa1
Reviewer #2 (Public Review): https://doi.org/10.7554/eLife.96190.3.sa2
Reviewer #3 (Public Review): https://doi.org/10.7554/eLife.96190.3.sa3
Author response https://doi.org/10.7554/eLife.96190.3.sa4

---

# Additional files

## Supplementary files
- Supplementary file 1. Reagents for multiplex fluorescence immunohistochemistry.
- MDAR checklist

## Data availability
All data generated or analysed during this study are included in the manuscript and supporting files.

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
