## [Editor Report · eLife assessment]

In this **useful** study, the authors report the efficacy, hematological effects, and inflammatory response of the BPaL regimen (containing bedaquiline, pretomanid, and linezolid) compared to a variation in which Linezolid is replaced with the preclinical development candidate spectinamide 1599, administered by inhalation in tuberculosis-infected mice. The authors provide **convincing** evidence that supports the replacement of Linezolid in the current standard of care for drug-resistant tuberculosis. The work will be of interest to those studying tuberculosis treatment regimens.

---

## [Referee Report · Reviewer #1 (Public Review)]

Summary:

The manuscript entitled A Modified BPaL Regimen for Tuberculosis Treatment

replaces Linezolid with Inhaled Spectinamides by Malik Zohaib Ali et al. is an extension of previous studies by this group looking at the new drug spectinamide 1599. The authors directly compare therapy with BPaL (bedaquiline, pretomanid, linezolid) to a therapy that substitutes spectinamide for linezolid (BPaS). The Spectinamide is given by aerosol exposure and the BPaS therapy is shown to be as effective as BPaL without adverse effects. The work is rigorously performed and analyses of the immune responses are consistent with curative therapy.

Strengths:

1. This group uses 2 different mouse models to show the effectiveness of the BPaS treatment.

Weaknesses:

1. Although this is not a weakness of this paper, a sentence describing how the spectinamide would be administered by aerosolization in humans would be welcomed.

---

## [Referee Report · Reviewer #2 (Public Review)]

Summary:

Replacing linezolid (L) with the preclinical development candidate spectinamide 1599, administered by inhalation, in the BPaL standard of care regimen achieves similar efficacy, reduces hematological changes and por-inflammatory responses.

Strengths:

The authors not only measure efficacy but also quantify histological changes, hematological responses and immune responses, to provide a comprehensive picture of treatment response and the benefits of the L to S substitution.

The authors generate all data in two mouse models of TB infection, each reproducing different aspects of human histopathology.

Extensive supplementary figures ensure transparency.

Weaknesses:

Articulation of objectives and hypotheses can be improved, as suggested below.

---

## [Referee Report · Reviewer #3 (Public Review)]

Summary:

In this paper, the authors sought to evaluate whether the novel TB drug candidate, spectinamide 1599 (S), given via inhalation to mouse TB models, and combined with the drugs B (bedaquiline) and Pa (pretomanid), would demonstrate similar efficacy to that of BPaL regimen (where L is linezolid). Because L is associated with adverse events when given to patients longterm, and one of those is associated with myelosuppression (bone marrow toxicity) the authors also sought to assess blood parameters, effects on bone marrow, immune parameters/cell effects following treatment of mice with BPaS and BPaL. They conclude that BPaL and BPaS have equivalent efficacy in both TB models used and that BPaL resulted in weight loss and anemia (whereas BPaS did not) under the conditions tested, as well as effects on bone marrow.

Strengths:

The authors used two mouse models of TB that are representative of different aspects of TB in patients (which they describe well), intending to present a fuller picture of the activity of the tested drug combinations. They conducted a large body of work in these infected mice to evaluate efficacy and also to survey a wide range of parameters that could inform the effect of the treatments on bone marrow and on the immune system. The inclusion of BPa controls (in most studies) and also untreated groups led to a large amount of useful data that has been collected for the mouse models per se (untreated) as well as for BPa - in addition to the BPaS and BPaL combinations which are of particular interest to the authors. Many of these findings related to BPa, BPaL, untreated groups etc corroborate earlier findings and the authors point this out effectively and clearly in their manuscript. To go further, in general, it is a well written and cited article with an informative introduction.

Weaknesses:

The authors performed a large amount of work with the drugs given at the doses and dosing intervals stated, but there is no exposure data available at this time. The authors intend to evaluate exposure-effect relationships in future work. An understanding of the exposures at which the efficacy and adverse effects are seen will assist in the translation of these findings to the clinic.

In addition, it is always challenging to interpret findings for combinations of drugs and for now, the data available cannot attribute confidence to the weight loss seen for only the BPaL group to L specifically, as opposed to a PK interaction leading to an elevated exposure and weight loss due to B or Pa. It is not yet possible, then to state that what is seen are "L-associated AEs" - this is assumed only.

The evaluations of activity in the BALB/c mouse model as well as the spleens of the Kramnik model resulted in CFU below/at the limit of detection so comparisons between BPaL and BPaS cannot be made and so the conclusion of equivalent efficacy in BALB/c is not supported with the data shown. There is no BPa control in the BALB/c study, therefore it is not possible to discern whether L or S contributed to the activity of BPaL or BPaS. The same is true for the assessment of lesions - unfortunately, there was no BPa control meaning that even where equivalency is seen for BPaL and BPaS, the reader is unable to deduce whether L or S made a contribution to this activity.

Although these weaknesses limit what we can learn from the current body of data, the authors note that further studies will be done to increase understanding of the points above.

---

## [Author Response]

The following is the authors’ response to the original reviews.

**eLife assessment**
In this useful study, the authors report the efficacy, hematological effects, and inflammatory response of the BPaL regimen (containing bedaquiline, pretomanid, and linezolid) compared to a variation in which Linezolid is replaced with the preclinical development candidate spectinamide 1599, administered by inhalation in tuberculosis-infected mice. The authors provide convincing evidence that supports the replacement of Linezolid in the current standard of care for drug-resistant tuberculosis. However, a limitation of the work is the lack of control experiments with bedaquiline and pretomanid only, to further dissect the relevant contributions of linezolid and spectinamide in efficacy and adverse effects.

We acknowledge a limitation in our study due to lack of groups with monotherapy of bedaquiline and pretomanid however, similar studies to understand contribution of bedaquiline and pretomanid to the BPaL have been published already (references #4 and #60 in revised manuscript). Our goal was to compare the BPaS versus the BPaL with the understanding that TB treatment requires multidrug therapy. We omitted monotherapy groups to reduce complexity of the studies because the multidrug groups require very large number of animals with very intensive and complex dosing schedules. Even if B or Pa by themselves have better efficacy than the BPa or BPaL combination, patients will not be treated with only B or Pa because of very high risk of developing drug resistance to B or/and PA. If drug resistance is developed for B or Pa, the field will lose very effective drugs against TB.

Although the manuscript is well written overall, a re-formulation of some of the stated hypotheses and conclusions, as well as the addition of text to contextualize translatability, would improve value.

Manuscript has been edited to address these critiques. Answers to individual critiques are below.

**Public Reviews:**

**Reviewer #1 (Public Review):**
Summary:This manuscript is an extension of previous studies by this group looking at the new drug spectinamide 1599. The authors directly compare therapy with BPaL (bedaquiline, pretomanid, linezolid) to a therapy that substitutes spectinamide for linezolid (BPaS). The Spectinamide is given by aerosol exposure and the BPaS therapy is shown to be as effective as BPaL without adverse effects. The work is rigorously performed and analyses of the immune responses are consistent with curative therapy.Strengths:(1) This group uses 2 different mouse models to show the effectiveness of the BPaS treatment.(2) Impressively the group demonstrates immunological correlates associated with Mtb cure with the BPaS therapy.(3) Linezolid is known to inhibit ribosomes and mitochondria whereas spectinaminde does not. The authors clearly demonstrate the lack of adverse effects of BPaS compared to BPaL.Weaknesses:(1) Although this is not a weakness of this paper, a sentence describing how the spectinamide would be administered by aerosolization in humans would be welcomed.

We already reported on the aerodynamic properties of dry powder spectinamide 1599 within #3 HPMC capsules and its delivery from a RS01 Plastiape inhaler device (reference #59 in revised manuscript). To address this critique, we added a last paragraph in discussion “It is proposed that human use of spectinamides 1599 will be administered using a dry powder formulation delivered by the RS01 Plastiape dry powder inhaler" (reference #59 in revised manuscript).

**Reviewer #2 (Public Review):**
Summary:Replacing linezolid (L) with the preclinical development candidate spectinamide 1599, administered by inhalation, in the BPaL standard of care regimen achieves similar efficacy, and reduces hematological changes and proinflammatory responses.Strengths:The authors not only measure efficacy but also quantify histological changes, hematological responses, and immune responses, to provide a comprehensive picture of treatment response and the benefits of the L to S substitution.The authors generate all data in two mouse models of TB infection, each reproducing different aspects of human histopathology.Extensive supplementary figures ensure transparency.Weaknesses:The articulation of objectives and hypotheses could be improved.

We edited to "The AEs were associated with the long-term administration of the protein synthesis inhibitor linezolid. Spectinamide 1599 (S) is also a protein synthesis inhibitor of *Mycobacterium tuberculosis* with an excellent safety profile, but which lacks oral bioavailability. Here, we propose to replace L in the BPaL regimen with spectinamide administered via inhalation and we demonstrate that inhaled spectinamide 1599, combined with BPa ––BPaS regimen––has similar efficacy to that of BPaL regimen while simultaneously avoiding the L-associated AEs.

**Reviewer #3 (Public Review):**
Summary:In this paper, the authors sought to evaluate whether the novel TB drug candidate, spectinamide 1599 (S), given via inhalation to mouse TB models, and combined with the drugs B (bedaquiline) and Pa (pretomanid), would demonstrate similar efficacy to that of BPaL regimen (where L is linezolid). Because L is associated with adverse events when given to patients long-term, and one of those is associated with myelosuppression (bone marrow toxicity) the authors also sought to assess blood parameters, effects on bone marrow, immune parameters/cell effects following treatment of mice with BPaS and BPaL. They conclude that BPaL and BPaS have equivalent efficacy in both TB models used and that BPaL resulted in weight loss and anemia (whereas BPaL did not) under the conditions tested, as well as effects on bone marrow.Strengths:The authors used two mouse models of TB that are representative of different aspects of TB in patients (which they describe well), intending to present a fuller picture of the activity of the tested drug combinations. They conducted a large body of work in these infected mice to evaluate efficacy and also to survey a wide range of parameters that could inform the effect of the treatments on bone marrow and on the immune system. The inclusion of BPa controls (in most studies) and also untreated groups led to a large amount of useful data that has been collected for the mouse models per se (untreated) as well as for BPa - in addition to the BPaS and BPaL combinations which are of particular interest to the authors. Many of these findings related to BPa, BPaL, untreated groups, etc corroborate earlier findings and the authors point this out effectively and clearly in their manuscript. To go further, in general, it is a well-written and cited article with an informative introduction.Weaknesses:The authors performed a large amount of work with the drugs given at the doses and dosing intervals started, but at present, there is no exposure data available in the paper. It would be of great value to understand the exposures achieved in plasma at least (and in the lung if more relevant for S) in order to better understand how these relate to clinical exposures that are observed at marketed doses for B, Pa, and L as well as to understand the exposure achieved at the doses being evaluated for S. If available as historical data this could be included/cited. Considering the great attempts made to evaluate parameters that are relevant to clinical adverse events, it would add value to understand what exposures of drug effects such as anemia, weight loss, and bone marrow effects, are being observed. It would also be of value to add an assessment of whether the weight loss, anemia, or bone marrow effects observed for BPaL are considered adverse, and the extent to which we can translate these effects from mouse to patient (i.e. what are the limitations of these assessments made in a mouse study?). For example, is the small weight loss seen as significant, or is it reversible? Is the magnitude of the changes in blood parameters similar to the parameters seen in patients given L? In addition, it is always challenging to interpret findings for combinations of drugs, so the addition of language to explain this would add value: for example, how confident can we be that the weight loss seen for only the BPaL group is due to L as opposed to a PK interaction leading to an elevated exposure and weight loss due to B or Pa?

We totally agree with this critique but the studies suggested by the reviewer are very expensive and

logistically/resource intensive. Data reported in this manuscript was used as preliminary data in a RO1 application to NIH-NIAID that included studies proposed above by this reviewer. The authors are glad to report that the application got a fundable score and is currently under consideration for funding by NIH-NIAID. The summary of proposed future studies is included in the last paragraph of the discussion in this revised manuscript.

Turning to the evaluations of activity in mouse TB models, unfortunately, the evaluations of activity in the BALB/c mouse model as well as the spleens of the Kramnik model resulted in CFU below/at the limit of detection and so, to this reviewer's understanding of the data, comparisons between BPaL and BPaS cannot be made and so the conclusion of equivalent efficacy in BALB/c is not supported with the data shown. There is no BPa control in the BALB/c study, therefore it is not possible to discern whether L or S contributed to the activity of BPaL or BPaS; it is possible that BPa would have shown the same efficacy as the 3 drug combinations. It would be valuable to conduct a study including a BPa control and with a shorter treatment time to allow comparison of BPa, BPaS, and BPaL.

We agree with the reviewer these studies need to be done. Some of them were recently published by our colleague Dr. Lyons (reference #60 in revised manuscript). The studies proposed by the reviewer will be performed under a new award under consideration for funding by the NIH-NIAID, the summary of future studies is included in the last paragraph of the discussion in this revised manuscript.

In the Kramnik lungs, as the authors rightly note, the studies do not support any contribution of S or L to BPa - i.e. the activity observed for BPa, BPaL, and BPaS did not significantly differ. Although the conclusions note equivalency of BPaL and BPaS, which is correct, it would be helpful to also include BPa in this statement;

We edited and now included in lines #191 as requested

It would be useful to conduct a study dosing for a longer period of time or assessing a relapse endpoint, where it is possible that a contribution of L and/or S may be seen - thus making a stronger argument for S contributing an equivalent efficacy to L. The same is true for the assessment of lesions - unfortunately, there was no BPa control meaning that even where equivalency is seen for BPaL and BPaS, the reader is unable to deduce whether L or S made a contribution to this activity.

Added in the future plans in the last paragraph of discussion

“Future studies are already under consideration for funding by NIH-NIAID to understand the pharmacokinetics of mono, binary and ternary combinations of BPaS. These studies also aim to identify the optimal dose level and dosing frequency of each regimen along with their efficacy and relapse free-sterilization potential. Studies are also planned using a model-based pharmacokinetic-pharmacodynamic (PKPD) framework, guided by an existing human BPa PKPD model (reference #61 in revised manuscript), to find allometric human dose levels, dosing frequencies and treatment durations that will inform the experimental design of future clinical studies.

**Recommendations for the authors:**

**Reviewer #1 (Recommendations For The Authors):**
Although this is not a weakness of this paper, a sentence describing how the spectinamide would be administered by aerosolization in humans would be welcomed.

Last paragraph of discussion was added “It is proposed that human use of spectinamides 1599 will be administered using a dry powder formulation delivered by the RS01 Plastiape dry powder inhaler". We already reported on the aerodynamic properties of dry powder spectinamide 1599 within #3 HPMC capsules and delivered from a RS01 Plastiape inhaler device (reference #59 in revised manuscript)

**Reviewer #2 (Recommendations For The Authors):**
Major commentsThe Abstract lacks focus and could more clearly convey the key messages.

Edited as requested

The two mouse models and why they were chosen need to be described earlier. Currently, it's covered in the first section of the Discussion, but the reader needs to understand the utility of each model in answering the questions at hand before the first results are described, either in the introduction or in the opening section of the results.

Thank you for suggestion, we agree. We moved the first paragraph in discussion to last paragraph in Introduction.

Line 130: Please justify the doses and dosing frequency for S. A reference to a published manuscript could suffice if compelling.

The dosing and regimens were previously reported by our groups in ref 21 and 22 in revised manuscript.-

(21) Robertson GT, Scherman MS, Bruhn DF, Liu J, Hastings C, McNeil MR, et al. Spectinamides are effective partner agents for the treatment of tuberculosis in multiple mouse infection models. J Antimicrob Chemother.

2017;72(3):770–7.

(22) Gonzalez-Juarrero M, Lukka PB, Wagh S, Walz A, Arab J, Pearce C, et al. Preclinical Evaluation of Inhalational Spectinamide-1599 Therapy against Tuberculosis. ACS Infect Dis. 2021;7(10):2850–63.

Figures 1 E to H: several "ns" are missing, please add them.

Edited as requested

Line 184 to 190: suggest moving the body weight plots to a Supplemental Figure, and at least double the size of the histology images to convey the message of lines 192-203.Please include higher magnification insets to illustrate the histopathological findings. In that same section, please add a sentence or two describing the lesion scoring concept/method. It is a nice added feature, not widespread in the field, and deserves a brief description.

Edited as requested. We added detailed description for scoring method in M&M under histopathology and lesion scoring

Line 206: please add an introductory sentence explaining why one would expect S to cause (or not) hematological disruption, and why MCHC and RDW were chosen initially (they are markers of xyz). The first part of Figure 3 legend belongs to the Methods.

To address this critique we added in #225-226 “The effect of L in the blood profile of humans and mouse has been reported (references #38-42 in revised manuscript) but the same has not been reported for S” . In line #229-230 we added “Of 20-blood parameters evaluated, two blood parameters were affected during treatment”.

The first part of Figure 3 legend belongs to the Methods.

We edited Figure 3 to “During therapy of mice in Figure 1, the blood was collected at 1, 2- and 4-weeks posttreatment. The complete blood count was collected in VETSCAN HM5 hematology analyzer (Zoetis)”.

Line 218: please explain why the 4 blood parameters that are shown were selected, out of the 20 parameters surveyed.

We added an explanation in line 239-240 “out 20-blood parameters evaluated, a total of four blood parameters were affected at 2 and 4-weeks-of treatment”.

Line 243 and again Line 262 (similar to comment Line 206): please add an introductory paragraph explaining the motivation to conduct this analysis and the objective. Can the authors put the experiment in the context of their hypothesis?

To address this critique, we added in line #235-237 “The Nix-TB trial associated the long-term administration of L within the BPaL regimen as the causative agent resulting in anemia in patients treated with the BPaL regimen (5).”

Figure 4C (and the plasma and lung equivalent in the SI). This figure needs adequate labeling of axes: X axis = LOG CFU? Please add tick marks for all plots since log CFU is only shown for the bottom line. Y axes have no units: pg/mL as in B?

Figure legend were edited to add (Y axis:pg/ml) and (X axis; log10CFU).

Line 255-256: please remove "pronounced" and "profound". There is a range of CFU reduction and cytokine reduction, from minor to major. The correlation trend is clear and those words are not needed.

Edited as requested

Line 277-289, Figure 6: given the heterogeneity of a C3HeB/FeJ mouse lung (TB infected), and the very heterogeneous cell population distribution in these lungs (Fig. 6A), the validity of whole lung analysis on 2 or 3 mice (the legend should state what 1, 2 and 3 means, individual mice?) is put into question. "F4/80+ cells were observed significantly higher in BPaS compared to UnRx control": Figure S14 suggests a statistically significant difference, but nothing is said about the other cell type, which appears just as much reduced in BPaS compared to UnRx as F4/80+. Overall, sampling the whole lung for these analyses should be mentioned as a limitation in the Discussion.

We agree with the reviewer that "visually" it appears as other populations in addition to F4/80 have statistical significance. We run again the two way Anova with Tukey test and only the BPaS and UnRx for F4/80 is significant.

We edited figure S16 (previously S14) to add ns for every comparation.

In Figure 6A was edited ; N=2 are 2 mice for Unrx and n=3 mice for BPaL/BPaS each.

Line 355-360: "The BPa and BPaL regimens altered M:E in the C3HeB/FeJ TB model by suppressing myeloid and inducing erythroid lineages" This suggests that altered M:E is not associated with L, putting into question the comparison between BPaS, BPaL, and UnRx. Can the authors comment on how M:E is altered in BPa and not in BPaS?

Our interpretation to this result was that addition of S in our regimen BPsS was capable of restoring the M:E ratio altered by the BPa and BPaL. This interpretation was included in main text in line #263-264 and is also now added to abstract

Line 379: discuss the limitations of working with whole lungs.

Sorry we cannot understand this request. In our studies we always work with whole lungs if the expected course of histopathology/infection among lung lobes is very variable (as is the case of C3HeB/Fej TB model)

Concluding paragraph: "Here we present initial results that are in line with these goals." If such a bold claim is made, there needs to be a discussion on the translatability of the route of administration and the dose of S. Otherwise, please rephrase.

We added the following last paragraph to discussion:

To conclude, the TB drug development field is working towards developing shorter and safer therapies with a common goal of developing new multidrug regimens of low pill burden that are accessible to patients, of short duration (ideally 2-3 months) and consist of 3-4 drugs of novel mode-of-action with proven efficacy, safety, and limited toxicity. Here we present initial results for new multidrug regimens containing inhaled spectinamide 1599 that are in line with these goals. It is proposed that human use of spectinamides 1599 will be administered using a dry powder formulation delivered by the RS01 Plastiape dry powder inhaler. We already reported on the aerodynamic properties of dry powder spectinamide 1599 within #3 HPMC capsules and delivered from a RS01 Plastiape inhaler device (reference #59 in revised manuscript). Future studies are already under consideration for funding by NIHNIAID to understand the pharmacokinetics of mono, binary and ternary combinations of BPaS. These studies also aim to identify the optimal dose level and dosing frequency of each regimen along with their efficacy and relapse free-sterilization potential. Studies are also planned using a model-based pharmacokinetic-pharmacodynamic (PKPD) framework, guided by an existing human BPa PKPD model (references #60 and 61 in revised manuscript) , to find allometric human dose levels, dosing frequencies and treatment durations that will inform the experimental design of future clinical studies.

Minor editsAdverse events, not adverse effects (side effects)

Edited as requested

BALB/c (not Balb/c, please change throughout).

Edited as requested

Line 92: replace 'efficacy' with potency or activity.

Edited as requested

"Live" body weight: how is that different from "body weight"? Suggest deleting "live" throughout, or replace with "longitudinally recorded" if that's what is meant, although this is generally implied.

Edited as requested

The last line of Figure 2 legend is disconnected.Line 331: delete "human".

Edited as requested

**Reviewer #3 (Recommendations For The Authors):**

We thank the reviewer for these suggestions. The data presented in this manuscript with 4 weeks of treatment along with monitoring of effects of therapy in blood, bone marrow and immunity have been submitted for a RO1 application to NIH-NIAID, which have received a fundable score and is under funding consideration. All the points suggested by the reviewer(s) here are included in the research proposed in the RO1 application including manufacturing and physico-chemically characterize larger scale of dry powders of spectinmides and evaluation of their aerodynamic performance for human or animal use; Pharmacokinetics and efficacy studies to determine the optimal dose level and dosing frequency for new multidrug regimens containing spectinamides. These studies include mono, binary and ternary combinations of each multidrug regimen along with their efficacy and relapse free- sterilization potential. These studies will also develop PK/PD simulation-based allometric scaling to aid in human dose projections inhalation. We hope the reviewer will understand all together these studies will last 4-5 years.

Although I truly appreciate the great efforts of the authors, I suggest that in order to better evaluate the contribution of S versus L to BPa in these models, repeat studies be run that:(a) include BPa groups to allow the contribution of S and L to be assessed. Included in research proposed RO1 application mentioned above(b) use shorter treatment times in BALB/c to allow comparisons at end of Tx CFU above the LOD. We have added new data for 2 weeks treatment with BPaL and BPaS in Balb/c mice infected with MTb that was removed from previous submission of this manuscript(c) use longer treatment times and ideally a relapse endpoint in Kramnik to allowassessment of L and S as contributors to BPa (i.e. give a chance to see better efficacy of BPaL or BPaS versus BPa) and also measure plasma exposures of all drugs (or lung levels if this is the translatable parameter for S) to allow detection of any large DDI and also understand the translation to the clinic. Related to the safety parameters, it would be really great to understand whether or not the observations for BPaL would be labeled adverse in a toxicology study/in a clinical study, and it would be useful to include information on the magnitude of observations seen here versus in the clinic (eg for the hematological parameters).

The research proposed in the RO1 application mentioned above included extensive PK, extended periods of treatment beyond 1 month of treatment (2-5 months as needed to reach negative culturable bacterial from organs) and of course relapse studies.

Minor point: I suggest rewording "high safety profile" when describing spectinomides in the intro - or perhaps qualify the length of dosing where the drug is well tolerated

"high safety profile" was replaced by “an acceptable safety profile”